



# Rapid Evolution of Aerosol Particles and their Optical Properties Downwind of Wildfires in the Western U.S.

Lawrence I. Kleinman[1], Arthur J. Sedlacek III[1], Kouji Adachi[2], Peter R. Buseck[3], Sonya Collier[4,a], Manvendra K. Dubey[5], Anna L. Hodshire[6], Ernie Lewis[1], Timothy B. Onasch[7], Jeffery R. Pierce[6], John Shilling[8], Stephen R. Springston[1], Jian Wang[1,b], Qi Zhang[4,b], Shan Zhou[4,c], and Robert J. Yokelson[9]

[1] Environmental and Climate Sciences Department, Brookhaven National Laboratory, Upton, NY USA

[2] Atmospheric Environment and Applied Meteorology Research Department, Meteorological Research Institute, Tsukuba, Japan

[3] School of Earth and Space Exploration, Arizona State University, Tempe, AZ USA

[4] Department of Environmental Toxicology, University of California, Davis, CA USA

[5] Earth Systems Observations, Los Alamos National Laboratory, Los Alamos, NM USA

[6] Department of Atmospheric Science, Colorado State University, Fort Collins, CO USA

[7] Aerodyne Research Inc., Billerica, MA USA

[8] Atmospheric Sciences and Global Change Division, Pacific Northwest National Laboratory, Richland, WA USA

[9] Department of Chemistry and Biochemistry, University of Montana, Missoula, MT 59812

a Now at California Air Resources Board
b now at Center for Aerosol Science and Engineering, Washington University, St. Louis, MO
c now at Department of Chemistry, Syracuse University, Syracuse, NY

*Correspondence to* Lawrence Kleinman (kleinman@bnl.gov)



**Abstract**

During the first phase of the Biomass Burn Operational Project (BBOP) field campaign, conducted in the Pacific Northwest, the DOE G-1 aircraft was used to follow the time evolution of wildfire smoke from near the point of emission to locations 2 – 3.5 hours downwind. In nine flights we made repeated transects of wildfire plumes at varying downwind distances and could thereby follow the plume's time evolution. On average there was little change in dilution-normalized aerosol mass concentration as a function of downwind distance. This consistency hides a dynamic system in which primary aerosol particles are evaporating and secondary ones condensing. Organic aerosol is oxidized as a result. On all transect more than 90% of aerosol is organic. In freshly emitted smoke aerosol, $NH_4^+$ is approximately equivalent to $NO_3^-$. After two hours of daytime aging, $NH_4^+$ increased and is approximately equivalent to the sum of $Cl^-$, $SO_4^{2-}$ and $NO_3^-$. Particle size increased with downwind distance causing particles to be more efficient scatters. Averaged over nine flights, mass scattering efficiency increased in ~ two hours by 56% and in one fight doubled. Coagulation and material transport from small to large particles are discussed as mechanisms for increasing particle size. As absorption remained nearly constant with age the time evolution of single scatter albedo was controlled by age-dependent scattering. Near-fire aerosol had a single scatter albedo (SSA) of 0.8 – 0.9. After one to two hours of aging SSAs were typically 0.9 and greater. Assuming global-average surface and atmospheric conditions, the observed age-dependence in SSA would change the direct radiative effect of a wildfire plume from near zero near the fire to a cooling effect downwind.

## 1. Introduction

Aerosols from wildfires alter Earths radiation balance by their direct interaction with sunlight and by indirect effects mediated through perturbations on clouds and precipitation. In the direct effect, heating is the result of light absorption by black carbon (BC), brown carbon (BrC) - including tar balls, and dust. Cooling occurs when aerosol particles scatter light upwards so that sunlight that would otherwise be absorbed by Earth or within the atmosphere, escapes to space. The transition between heating and cooling depends on intrinsic aerosol properties such as aerosol single scatter albedo (SSA= light scattering/light extinction) and extrinsic factors, including surface albedo, solar zenith angle, and cloud cover (Chylek and Wong, 1995; Nemesure and Schwartz, 1995; McComiskey et al., 2008).

On a global basis, wildfire aerosols are estimated to have a net direct radiative effect of 0.17 watts $m^{-2}$, 1-σ uncertainty = -0.45 - +0.15 W $m^{-2}$ (Bond et al., 2013) The rapid evolution of aerosol optical properties of fresh smoke contributes to the radiative effect uncertainty (Yokelson et al., 2009; Akagi et al., 2012; May et al., 2015; Vakkari et al., 2018; Selimovic et al., 2019). In this study we are concerned with changes in wildfire optical properties that occur in the near-field, extending from the time it takes a smoke plume to rise to aircraft-sampling altitude to circa three hours downwind. These first hours are a dynamic period as the plume's initial store of reactive VOCs and $NO_x$ are largely intact and can generate secondary





low volatility species or species with light absorbing functional groups. Aerosol particles respond to
dilution and to gas and condensed phase chemical reactions. Particle size and composition change, altering
optical properties. Radiative effects are generally expected to follow trends in optical parameters, though
there is no simple one-to-one correspondence.

Fuel type and burn conditions vary from region to region causing wildfire aerosol to have varied
properties (Akagi et al., 2011; Andreae, 2019). Tropical regions have received attention because of the
amount of burning, and for the practical reason that there is often large-scale burning during a predictable
dry season. Temperate regions, in particular within the United States, have until recently received less
attention because locations and times of wildfires are less predictable. One focus of biomass burning (BB)
studies has been the evolution of organic aerosol downwind of wildfires and prescribed burns; summaries
of which are given by Garofalo et al. (2019) and Hodshire et al. (2019a). A comparison of smoke plumes
near fires with those several hours downwind indicates that organic aerosol (normalized by CO to account
for plume dispersion) varies from fire to fire but on average there is little change in normalized aerosol
mass during the first several hours of transport. Increases, when observed have had maximum values near a
factor of two (Yokelson et al., 2009), much smaller than seen in urban areas (e.g. Weber et al., 2007;
Kleinman et al., 2008). More typical were fires observed in the WE-CAN campaign in which normalized
aerosol concentration were near constant (Garofalo et al., 2019). Laboratory and field measurements of
organic aerosol species indicative of oxidation state and volatility show that composition changes with time
and that the net change in organic aerosol is affected by loss due to evaporation and gain due to
condensation of secondary organic species (May et al., 2013, 2015; Morgan et al., 2019). Although no
fundamental reasons are known why evaporation and condensation have comparable magnitude, the
aircraft measurements from WE-CAN and other studies, including the BBOP field campaign presented
here, find that such cancellation can indeed occur.

Changes in aerosol size distribution with age occur downwind of wildfires and prescribed burns (e.g.
Janhäll et al., 2010; Liang et al., 2016). Ultrafine particles (Dp = 3 to 10 nm diameter) are prevalent near
fires and rapidly removed by coagulation (Carrico et al., 2016; Sakamoto et al., 2016). Evaporation of
volatile species reduces particle size whilst coagulation and condensation of lower volatility species results
in particle growth. Coincident with this mass transfer are chemical reactions in the gas and aerosol phase
which, at least initially, proceed in the direction of creating less volatile, more oxidized species. Even
though normalized aerosol mass concentrations may not change significantly with age, mass can be
transferred between different size particles through coagulation and particle-vapor mass transfer. Our
interest in evolving size distributions is motivated by the dependence of light scattering on aerosol size.
Optically relevant BB aerosol typically have diameters between 100 nm and several hundred nm, a size





range over which light scattering per unit mass of aerosol (MSE = mass scattering efficiency) varies

several–fold.

The first phase of the Biomass Burning Operational Project (BBOP; Kleinman and Sedlacek, 2016) field campaign was conducted in the temperate Pacific Northwest. Twenty-one research flights were conducted between July and September 2013. Primary targets were i) wildfires in which the time evolution

of aerosols and trace gasses could be determined from measurements at multiple distances between the fire and locations several hours (smoke physical age) downwind and ii) overflights of a ground site at Mount Bachelor Observatory (MBO) in Oregon and areas upwind at times when MBO was impacted by wildfire smoke (Collier et al., 2016; Zhou et al., 2017; Zhang et al., 2018). One MBO flight coincided with a SEAC[4]RS flight (Liu et al., 2017). In the second phase of BBOP, during October 2013, agricultural burns

in the lower Mississippi Valley were sampled. In both phases of BBOP, TEM samples were analyzed for tar balls (TBs) as reported by Sedlacek et al. (2018a) and Adachi et al (2018; 2019).

In this study we use data from five wildfires that were collected during nine flights. Pseudo-Lagrangian sampling allows us to determine the rate of change of aerosol, gaseous, and optical quantities as

a function of transport time or photochemical age. Extensive variables are normalized to CO to account for dilution. We use the negative logarithm of the ratio of $NO_x$ to $NO_y$ ($-\mathrm{Log}_{10}(NO_x/NO_y)$) as a metric for photochemical processing. Of primary interest are time dependent changes in the concentration of sub-micrometer diameter aerosol, mass scattering efficiency (MSE), and mass absorption coefficient (MAC). The latter two quantities yield a local determination of SSA. Though there was only a small effect of aging

upon total aerosol mass concentration, within a couple of hours there were i) changes in the concentrations of inorganic species and in organic speciation as judged by atomic ratios and mass peaks at m/z = 43, 44, and 60 and ii) a significant increase in scattering and MSE causing plumes to transition from a near-neutral radiative forcing to one that is cooling. Because observed size distributions were limited to Dp < 210 or 260 nm and most scattering is from larger particles we cannot (without extrapolating the size data)

calculate MSE for comparison with the observed ratio of scattering to aerosol mass. Instead, we have determined the diameter within the accumulation mode where the number size distribution has a peak and using that diameter as a metric for particle size we show that MSE is positively correlated with particle size and negatively correlated with backscatter ratio (total scattering/scattering into rear hemisphere) and scattering Angstrom exponent. These correlations between MSE and particle size and between MSE and

scattering properties give support to the hypothesis that MSE increases with age because coagulation and/or chemical and microphysical processes shift particle mass from small diameters to the large diameters where scattering is more efficient.

## 2. Experimental methods



In the first phase of BBOP the DOE G-1 aircraft sampled wildfire plumes in the temperate forests and range lands of the western U.S., focusing on time evolution from near a fire to a few hours transport time downwind.  An enhanced suite of instruments at Mount Bachelor Observatory (MBO), Oregon provided a continuous record of BB plume properties.  In coordinated flights the G-1 sampled plumes that later impacted MBO (Collier et al., 2016; Zhou et al., 2017) and one flight in Southern Oregon coordinated
with the SEAC[4]RS DC8 (Liu et al., 2017).

       An extended deployment, June 7, 2013 – Sept. 13, 2013, was made possible by basing the G-1 at the home of the ARM Aerial Facility (Schmid et al., 2014) in Pasco, WA.  The choice of Pasco as a center of operations was justified by examining alternate sites, covering most of the U.S. Western forest fire
region.  For each location the frequency of fires was determined based on daily emissions of $CO_2$, CO, NMHC, and PM2.5 from MODIS fire products and the FINNv1 (FIre INventory from NCAR) emission inventory (Wiedinmyer et al., 2011).  Annual variability was estimated from 10 years of data. Fire counts and emissions were summed over the G-1s 500 km operational range from the candidate sites.

**2.1. Instruments**

       Schmid et al (2014) review the G-1 aircraft and the instrument suite provided and maintained by the ARM Aircraft Facility. The following discussion and the list in Table 1 is restricted to instruments used in the present study. The G-1 instruments used in BBOP are also summarized by Liu et al. (2016).

***2.1.1 SP-AMS***

       The BBOP field campaign was among the first aerial deployments of the Soot Particle Aerosol Mass Spectrometer (SP-AMS). For an overview of the SP-AMS, see Onasch et al (2012).  In brief, the SP-AMS is a high-resolution time-of-flight aerosol mass spectrometer (HR-tof-AMS) to which a high-power CW YAG laser is added for vaporizing light absorbing refractory aerosol such as BC.  The SP-AMS can be
operated in laser-off or laser-on modes.  With the YAG laser off, non-refractory (nr) particles are vaporized by impacting a heated target, typically at 600 ℃; performance characteristics such as collection efficiency (CE) and fragmentation patterns are identical to an HR-tof-AMS.  In laser-on mode, strongly absorbing aerosol components such as BC are vaporized before encountering the heated target. Non-absorbing species that coat BC particles are vaporized by heat transfer as happens in an SP2 (Schwarz et al., 2010).  Brown
carbon (BrC), so-called because of its short-wavelength absorption, doesn't absorb the 1064 nm YAG laser unless there is a long wavelength absorption tail. The thermal vaporizer was left on when operating in laser-on mode, thereby allowing nr-aerosol to be quantified, but at the expense of not being able to selectively detect the coating on BC particles.

The SP-AMS and constant pressure inlet were operated as described by Collier et al., 2016.  One Hz data was acquired in "Fast-MS" V-mode.  Instrument-backgrounds were determined by alternating between





52 seconds of signal collection and 8 seconds with the aerosol beam blocked. Most SP-AMS data was acquired in laser-on mode. In order to investigate sampling strategies, we sometimes alternated between laser-on and laser-off operation, either minute by minute or for repeated plume crossings. The collection

efficiency of the SP-AMS during laser-off operation was determined to be 0.5 via comparisons between aerosol mass concentrations measured on the G-1 with similar measurements at MBO during overflights (see Fig. S2 of Collier et al., 2016). In each case we normalized non-refractory aerosol concentration to CO to account for changes in plume dilution between transects. The average ratio for 16 transects; laser-on to laser-off was 1.52. This procedure yielded a laser-on CE of 0.76 with a standard deviation of 0.07.


We assume, as done implicitly in other studies, that TBs contribute towards total non-refractory organics with the same detection efficiency as other organic aerosol. This assumption needs to be tested for ambient aerosol of the types observed in BBOP, especially since TBs were observed to have a mass fraction of 25% - 40% of PM1 in aged smoke (Sedlacek et al., 2018a). As a first step, the thermal stability

of TBs was investigated by electron microscopy of BBOP samples affixed to a heating stage in which the temperature was ramped up over 10 minutes to the 600 °C temperature of the SP-AMS thermal vaporizer (Adachi et al., 2018). At 600 °C, 30% of the TB mass collected on TEM grids was not vaporized. If the results of these slow-heating experiments are applicable to the thermal flash vaporization occurring in an AMS, a fraction of nominal nr-organic mass is not vaporized and therefore not detected. We estimate this

fraction as being of order 10% (25-40% mass fraction TB × 30% not evaporated in TEM experiments). Evidence to the contrary, namely, that TBs are detected with the same efficiency as other organic aerosol by an AMS is that TBs have the correct magnitude and volatility (up to 200 °C) to coincide with BBOA-3, a low volatility PMF factor of BB smoke sampled at MBO by Zhou et al. (2017).

Sedlacek et al. (2018b) and Corbin and Gysel-Beer (2019) showed that, depending on laser power, TBs can be charred by an SP2 YAG laser identical to that used in the SP-AMS. The charred residue has the potential to be detected as BC in a SP-AMS. BC mass concentrations in aged, temperate wildfire plumes were observed to be an order of magnitude less than TBs (Sedlacek et al., 2018a) and therefore BC concentrations are susceptible to large errors if even a small fraction of TB mass is mis-identified as BC. In

this study we use BC concentrations as observed by an SP2. In keeping with the nomenclature of the SP2 community, the light absorbing, incandescing species measured by the SP2 will be referred to as rBC.

In recent measurements at the USDA Fire Sciences Laboratory, Lim et al (2019) used an AMS to analyze BB generated aerosols soon after emission and then again after exposure to oxidants in a chamber

simulating up to several days of photochemical aging. The AMS collection efficiency (CE) was determined by comparison with SMPS measurements of aerosol size distribution, integrated to give a total volume. The AMS CE was 0.54 for fresh emissions, decreasing to 0.40 for multi-day aged aerosol. This decrease correlated with a decrease in aerosol volatility as measured by a thermal denuder. Aged particles





were interpreted as being more viscous and more likely to bounce off the AMS oven and thereby escape

detection.  If we assume the change in CE found by Lim et al (2019) our assumption of a constant CE
results is a 35% underestimate for the change in aerosol mass concentrations in aged air relative to fresh
emissions.

        Plume observations in BBOP covered a maximum aging time of 3.5 hours.  During aging we find,
as have others (e.g., DeCarlo et al., 2010; Heald et al., 2010; Ng et al., 2011) an increase in the aerosol O to
C ratio (O:C); a compositional change that has been observed to be correlated with increased viscosity.
This change in viscosity in aged smoke is consistent with the conclusion of Sedlacek et al (2018a) that
high-viscosity tar balls, found only after aging, are processed primary particles. The dependence of CE on
O:C has been determined for laboratory biomass burns by Onasch (personal communication).  Combined
with the O:C changes found in BBOP (presented below) we obtain an estimate that aged BBOP plumes can
contain up to 15% less aerosol mass than measured based on a fixed CE. We have not incorporated an age-
dependent CE into our calculations.  It is likely to be small for the range of aging that we cover and to the
best of our knowledge AMS observations of ambient biomass burn plumes have not shown systematic
changes in CE.


### 2.1.2 SP2

        A single particle soot photometer (SP2; Revision D), Droplet Measurement Technology) was used
to measure the number concentration and size distribution of refractory particles. Aerosol particles are
subjected to high intensity CW 1064 nm light from a YAG laser, heating particles that absorb at this
wavelength to the point of incandescence.  Color temperature of the incandescence is used to discriminate
between BC and other absorbers such as mineral dust.  The size distribution of rBC particles with volume
equivalent diameters between ~ 80 and 500 nm was determined based on a rBC density of 1.8 g cm$^{-3}$.
Fullerene soot (Alfa Aesar; stock no. 40971: lot no. L18U002) was used for calibration.

### 2.1.3 FIMS

        Aerosol size data used in this study are from the FIMS (Fast Integrated Mobility Spectrometer)
(Olfert et al., 2008; Wang et al., 2017), designed to provide information similar to an SMPS. Because the
FIMS measures particles of different sizes simultaneously instead of sequentially as in a traditional SMPS,
it provides aerosol size spectra with high time resolution (i.e. a size spectrum is acquired in 1 sec vs. 60
secs for the SMPS).  The minimum size particle classified had a diameter of 20 nm.  The classifying
voltage in the FIMS is reduced at high altitude to prevent arcing.  At the altitudes where plumes were
sampled (typically 2500 – 3000 m above msl) the upper size limit of the FIMS deployed during BBOP was
limited to 210 nm or 260 nm as a result of the reduced classifying voltage.  The FIMS size range usually
encompassed the peak of the particle number size distribution, dN/dLogDp.  FIMS data near the peak in
dN/dLogDp (restricted to the accumulation mode, between 100 nm and 260 nm) were used to estimate the





diameter (between bin boundaries) at which the peak occurs. We use this derived diameter as a surrogate for the geometric mean diameter, $D_{GEO}$, of the accumulation mode.

### 2.1.4 Scattering

Aerosol scattering, and back scatter ratio were determined from a three wavelength (450, 550, 700 nm) TSI 3563 nephelometer. Unless otherwise noted, scattering will refer to measurements at 550 nm. Flow rate and internal volume limited response time to 2 sec. In the G-1 data set, corrections for the deviation of the instrument from a cosine response and the smaller correction for not detecting photons scattered into a 7 degree forward and 10 degree backward cone were applied using the sub μm formulas

from Anderson and Ogren (1998). Size spectra up to 50 μm were determined from a Cloud, Aerosol, and Precipitation Spectrometer (CAPS) probe mounted outside the fuselage. The default CAPS diameter bins for liquid water were mapped onto bins corresponding to a refractive index of 1.55, as appropriate for an organic dominated aerosol. According to Mie calculations, supra-micrometer particles ($1\ \mu m < Dp < 5\ \mu m$) contributed only a couple of percent to total scattering. Nephelometer measurements were therefore

interpreted as being due to sub μm particles.

### 2.1.5 Light Absorption

    Aerosol light absorption at 532 nm was measured with a Photothermal Interferometer (PTI; Sedlacek and Lee, 2007). This instrument relies on an interferometer to detect a change in optical path

length caused by aerosol light absorption and subsequent degradation of absorbed energy into heat. As such the PTI is not susceptible to filter based artifacts affecting the PSAP and similar devices (Lack et al., 2008). For the calculation of single scatter albedo (SSA) we assume that absorption varies as $1/\lambda$, as is approximately correct for BC. PTI measurements are thereby decreased by 3% in order to adjust absorption measurements to 550 nm, the wavelength at which scattering data is available. Based on the 3-

λPSAP, it is likely that there is some BrC absorption at this wavelength but in view of the smallness of the adjustment it is neglected.

### 2.1.6 Trace Gases

    NO, $NO_2$, and $NO_y$ were measured in a 3-channel chemiluminescence instrument (Air Quality

Design, Golden, CO). Ambient NO was measured in the first channel with a small delay volume to give simultaneity with the $NO_2$ channel. $NO_2$ conversion occurs in a low-pressure cell with LED irradiation at 390 nm. Conversion efficiency for $NO_2$ was measured to be 0.50 +/- 0.03. $NO_y$ (defined as odd-nitrogen species including aerosol nitrate) was measured via a Mo converter heated to $350^oC$ located externally on a pylon affixed to a window blank. Conversion efficiency for $NO_2$ was 0.98 +/- 0.02 and earlier tests have

shown similar efficiencies for $HNO_3$ and organic nitrates (Williams et al., 1998).





CO, N$_2$O, and H$_2$O were measured with a commercial analyzer (Los Gatos Research, San Jose, CA) based on Off-Axis Integrated Cavity Output Spectroscopy (OA-ICOS). Standard additions were performed in flight and confirmed an accuracy of 1-2%. Precision at ambient backgrounds of 90 ppb CO was ~0.5

ppbv RMS at 1-s averaging (the internal volume of the instrument limited actual resolution to ~7-10 s). CO$_2$ was measured with a Picarro G1301-m cavity ring down spectrometer. Standard additions were performed in flight. Measurements of excess CO$_2$ in plumes was typically limited by the natural variability in background CO$_2$.

**2.1.7 Coincidence, Flow Control, and Dilution system**

It was anticipated that aerosol number concentrations close to wildfires would exceed the coincidence thresholds of several particle-resolved instruments (Table 1). At threshold a specified small percentage (e.g. 10%) of aerosol particles cannot be distinguished one from the next, resulting in missed counts or particles sorted into incorrect size bins. To mitigate coincidence errors, zero-air from a cylinder

was used to dilute ambient air in a ratio that was as high as 10 to 1. Dilution rates were changed in-flight to maintain a high instrument sensitivity in clean air. CPC data were corrected for coincidence.

**2.2 Data**

Concentrations and optical measurements are reported at STP, 0 °C and 1 atm. pressure. Particles

sizes are diameters. For each instrument and each flight, one Hz data were time shifted to maximize the correlation with light scattering, a time standard selected because of its availability on all flights and because it exhibits high correlations with most other quantities. Time shifts account for instruments sampling the same air parcel at different times. At a sampling speed of 100 m s$^{-1}$, km-scale plumes appear in the data record as rapidly changing signals. Time-shifts of 1-2 seconds are readily apparent as a

degradation in correlation when comparing instruments. Maximizing correlations, however, does not accurately compensate for varying response times. In so far as possible, we rely on average values across a plume, which are relatively insensitive to time response.

Except as noted, aerosol concentrations are the mass of non-refractory components as measured by

the SP-AMS and are referred to as nr-PM1. Units are μg/m$^3$ at STP. By presenting nr-PM1 we do not limit the aerosol concentration measurements by the requirement that the SP2 was also acquiring data at the same time. Mass differences between non-refractory and total aerosol are ~ 0.5 – 2.5%. References to aerosol number concentration (particles/cm$^3$ at STP) will be explicit to distinguish from aerosol mass.

Flights are identified by month (m), day (dd), and an "a" or "b" for the 1$^{st}$ or 2$^{nd}$ flight of a day, e.g., 821b was the second flight on Aug. 21, 2013. All times are UTC. Local time was Pacific-daylight savings and is given by UTC – 08:00. Local noon over our sampling region on 1 August ranged from 19:43 UTC


in the east to 20:15 UTC in the west. Data from the BBOP field campaign have been archived at
([https://www.arm.gov /research/campaigns/aaf2013bbop](https://www.arm.gov/research/campaigns/aaf2013bbop).


### 2.3 Flights and Wildfires

Figure 1 is a composite ground track for the 21 BBOP research flights conducted in the Pacific
Northwest. A synopsis of these flights is given in Table 2. Smoke from near-by biomass burns (BBs) was
observed on 18 flights: 17 from wildfires and one from a prescribed agricultural burn. On three BB flights,
measurements were made upwind and over a surface site at MBO (Collier et al 2016; Zhou, 2017; Zhang et
al., 2018). Four flights were devoted to observing urban plumes from Portland, Seattle, and Spokane. Tar
ball studies based on BBOP data make use all Pacific Northwest flights as well as 11 flights from the
second phase of BBOP in which agricultural burns in the lower Mississippi River Valley were sampled.

Wildfire flights analyzed for aging in this study had to meet three criteria. 1) Sampling included
fresh emissions, estimated as having an atmospheric residence time less than 30 minutes. That emissions
were fresh was determined by the observation of a high concentration, compact plume with a high $NO_x$ to
$NO_y$ ratio. 2) Measurements were made at downwind distances where significant aging was expected.
Transport between regions with fresh and aged pollutants had to be consistent with in-plume wind
measurements. 3) Measurements of aerosol mass, light scattering, and CO were required.

Nine pseudo-Lagrangian flights sampled smoke from five wildfires (Tables 3 and 4) for which the
time evolution of smoke could be followed from near its source to locations several hours downwind.
Evidence that fire characteristics were reasonably steady – an implicit assumption upon which Lagrangian
analysis depends - comes from flights in which two sets of transects were repeated with about a one-hour
delay between samples at nearly the same location. True Lagrangian sampling was not possible. Leaving
aside the difficulty of identifying and following a particular air parcel, the time over which this could be
done is restricted by the G-1's limited sampling time of one to two hours on station. Ground tracks for
flights 730b and 821b are shown in Figs. 3 and 7, respectively. The other seven flights satisfying our
pseudo-Lagrangian criteria are shown in Figs. S1 – S7. For the nine-flight ensemble, plumes were sampled
at downwind distances between a few km and 90 km, at which point up to 3.5 hours of aging occurred.
The minimum downwind distance is dictated by the distance needed for the plume to reach aircraft altitude
or, in some cases, by flight restrictions. Seven of the nine flights had $NO_x$ and $NO_y$ measurements that
were used to calculate photochemical age.


Several flights resemble regular grids containing two sets of up to 6 cross-plume segments. The
spacings between segments represents 20 – 60 minutes of plume aging. Other flights are more free-form
and in one case (821b) there is an extended along-plume segment that has been split into pieces with
different ages. The ground track figures indicate the portions of each flight, 106 in total, over which





averages are taken. We will refer to these cross-plume and along-plume flight segments as "transects". Transects were defined to be in the wildfire plume at a relatively narrow range of distances from the parent wildfire. Vertical plume structure was not explored because to do so would have been at the expense of horizontal spatial coverage. Wildfire sampling was mostly at constant altitude, at least 1000m above terrain, and insofar as possible near the altitude with maximum concentrations. Unlike small prescribed

burns, or observations from a far-downwind vantage point, the wildfires sampled in BBOP have a significant spatial extent in comparison to our measurement domain. There were often hot spots (secondary fires) in nominally downwind locations. Thus, scatter in our measurements as a function of downwind aging is expected.

Data for the nine pseudo-Lagrangian flights comes from the five wildfires listed in Table 3. Descriptions of these fires are available on National Fire Service Incident reports (https://fam.nwcg.gov/fam-web/hist_209/hist_2013_r_209_gacc_sprd?v_gaid=NW. Emission factors for the Colokum Tarp flights (730a and 730b) are compared with SEAC[4]RS and other data sets by Liu et al (2016).


## 3. Methods

### 3.1 Time Evolution

     The time evolution of fire emissions is calculated from plume measurements at varying downwind distances. We use transect-averaged quantities from which backgrounds have been subtracted. In the

example of species X, an excess concentration is

$$\Delta X_i = X_i - X_B \tag{1}$$

$X_i$ is an average of X over transect i and $X_B$ an average over a relatively unpolluted background region near

the smoke plume. If Xi is an intensive variable, such as an AMS mass ratio (*f*43, *f*44, *f*60, O to C ratio, or H to C ratio), a scattering Angstrom exponent, or a backscatter ratio, transect-averages are constructed using nr-PM1 as a weighting function. Problems and alternate ways of determining background are discussed by (Yokelson et al., 2013; Briggs et al., 2016; Garofalo et al., 2019). To account for plume dilution or conversely sampling a more concentrated plume region downwind, we divide measured

concentrations by a conservative tracer, CO. A normalized excess concentration is given by

$$\Delta X_i / \Delta CO_i = (X_i - X_B) / (CO_i - CO_B) \tag{2}$$

A ratio of $X_i$ to another quantity, $Y_i$, is calculated as


$$\Delta X_i / \Delta Y_i = (X_i - X_B) / (Y_i - Y_B) \tag{3}$$



In any ratio, such as in Eqs. 2 and 3, a common set of data points are used for numerator and denominator. Except as noted, ratios involving aerosol mass are based on non-refractory aerosol measured with the SP-AMS, i.e. nr-PM1. Equations (1) to (3) apply to variables that are expressed as concentrations, mixing ratios, or inverse lengths. The meaning should be clear from the units used.


Effects of aging are determined from changes in normalized excess ratios as a function of photochemical age or transport time. We define photochemical age by the conversion of $NO_x$ to oxidation products, expressed as $- Log_{10} (\Delta NO_x/\Delta NO_y)$ (Olszyna et al., 1994; Kleinman et al., 2008; DeCarlo et al., 2010). In the case that $NO_y$ is conserved and $NO_x$ is lost primarily by $OH + NO_2 \rightarrow HNO_3$, photochemical age, so defined, would yield [OH]*time; and given a trajectory-based time, would yield an average OH concentration. We, however, refrain from inferring an OH concentration from $NO_x$ and $NO_y$ measurements. We observe an apparent loss of $NO_y$ in fresh plumes (e.g., Neuman et al., 2004). Oxidation of $NO_x$ is more rapid than expected from $OH+NO_2$ (Mebust et al., 2011) which can be due to the known high yields of PAN (Alvarado et al., 2010; Briggs et al., 2016; Liu et al., 2016). As will be shown, $-Log_{10} (\Delta NO_x/\Delta NO_y)$ is a useful metric for chemical processing as it varies in the same sense as age-related changes in organic aerosol speciation.



## 4. Results


Brief summaries of the 9 pseudo-Lagrangian flights are given in Table 4. Transect average modified combustion efficiency (MCE) observations for the 6 pseudo-Lagrangian flights with $CO_2$ observations are tightly grouped between 0.86 and 0.92, close to MCE=0.9, traditionally taken as a transition point between mostly burning and mostly smoldering fires (Akagi et al., 2011). The relation between MCE and aerosol composition for two regional BBOP flights has been discussed in conjunction with measurements at MBO by Collier et al 2016. Amongst the 9 flights, emission intensity as determined by peak values of nr-PM1, CO, and light scattering varied by about an order of magnitude. Three fires were sampled on multiple flights, the Mile Marker 28 fire (725a and 726a), Colockum Tarps fire (730a, 730b, and 809a) and the Pony Fire Complex (813a and 814a). Plumes from the two Pony Fire Complex flights tended to resemble each other and have a similar age dependence (see figures, below), more so than the Mile Marker 28 and Colockum Tarps flights. In the most aged transects, background was an appreciable fraction of plume values for nr-PM1, CO, and scattering. For several flights background subtraction was problematic for $CO_2$ and inorganic aerosol components.



### 4.1 Plume Age, Concentration, and Dilution


Figure 2 shows a comparison between $NO_x/NO_y$ based photochemical age and atmospheric transport times. Photochemical age increases with downwind distance, but these two metrics of atmospheric processing are not directly proportional, nor should they be. In older plumes, $NO_x$ is depleted, and photochemical age tends to level off whilst distance is not bounded. Close to the fire transects have an





age ranging from 0.1 to above 0.5. The higher age values are generally from smaller, perhaps more rapidly
evolving, fires located near the main source of smoke. In the absence of other information, these transects
are assigned a downwind distance equal to that of nearby less aged transects. $NO_y$ data is missing from the
809a and 813a flights. Photochemical ages for transects on these flights have been generated from
downwind travel times and the 7-flight fit shown in Fig. 2.


The time evolution of BB plume constituents is affected by plume dilution because processes such
as gas phase oxidation and the partitioning of POA and SOA between phases are concentration dependent
(Hodshire et al, 2019b). In order to provide context for the BBOP data set and to facilitate comparison with
other studies we present in Table 4 ambient temperatures, MCE, and peak near-fire values for scattering,

mixing ratio of CO and concentration of nr-PM1. A dilution rate for each flight was determined from a fit
to the peak mixing ratio of CO on each transect, plotted as a function of photochemical age in Fig. S8. Our
measures of dilution are only qualitative as no attempt was made to sample an entire plume cross section.
Within the time taken for photochemical age to change from 0.2 to 1.0, peak plume mixing ratios of CO
decreased on average by a factor of 4.3. If measurements had been made at lower altitude, starting closer

to the fire, dilution rates would have been much higher (Hodshire et al., 2019b).

*4.2 Case Studies*

*4.2.1 Flight 730b*

The ground track for flight 730b, shown in Fig. 3, consists of two nearly identical sets of transects at

6 downwind distances. Observation times for the second set follow the first set by about one hour. Highest
concentrations are on plume-crossing #1; there are short duration spikes with the ratio of $NO_x$ to $NO_y$
approaching one, and CPC25 concentrations above $10^6$ cm$^{-3}$. In Fig. 3, plume-crossing #2 appears similar
to #1. But on crossing #2 photochemical age has increased, and the CPCs show far fewer particles with
diameter below 10 nm. On that basis, it appears that the plume crossings at location #2 are downwind of

the main fire region. We cannot, however, dismiss the possibility that the active burning region extends to
nominal downwind transects, a consideration pertinent to other flights.

Time series data for nr-PM1, Cl⁻, CO, and scattering are displayed in Fig. 4. Peak heights of the
conservative tracer CO (and the non-conservative quantities, nr-PM1, Cl⁻, and scattering) decrease with

distance as the plume becomes wider and more extended in altitude. As downwind distance and
photochemical age increase, the traces in Fig. 4b diverge indicating that the ratio of scattering to CO
increases. An increase in scattering/CO and MSE with age is a general feature of the wildfires studied.

Transect averaged quantities were used to compare Sets 1 and 2. As examples we present in Fig. 5

photochemical age vs. downwind travel time and the ratio of scattering to nr-PM1 as a function of
photochemical age. Differences between Sets 1 and 2 are due to the precision of our measurements and





natural time changes in the fire  Variations between observations in Set 1 and Set 2 can also be seen in the spatial displacement of plumes between repeated transects (Fig. 3), though allowance should be made for repeated transects not being exactly coincident.


In basing our analysis of plume evolution on wide cross-plume averages we are ignoring smaller scale structure.  For flight 730b most transects encompass two fire regions separated by 10's of km. One fire region is to the north and one to the south of the dashed line in Fig. 3.  A plot of $Cl^-$ vs. nr-PM1 in Fig. 5 (or a comparison of plots in Fig. 4a) shows that aerosol from the south fire has a $Cl^-$/nr-PM1 fraction of ~

2.5%, compared with 0.3% in the north. At the more downwind transects these two plumes overlap giving intermediate ratios. Differences in $Cl^-$ fraction are due to fuel type with the higher $Cl^-$ fraction characteristic of grass-land fires (Stockwell et al., 2014).  Other fire properties vary between north and south.  Onasch et al (2018), have shown that the low $Cl^-$ plume has a higher O to C ratio (0.37 vs. 0.31). The Colockum Tarp fire on flights 730a and 809a and the Mile Marker 28 fire on Flights 725a and 726a

also had a bimodal $Cl^-$ to total nr-aerosol ratio.

### 4.2.2 Flight 821b

Concentrations of gasses and aerosols sampled from the Government Flats fire on the 2nd August 21 flight were a factor of two higher than observed in other wildfires.  The ground track, shown in Fig. 7 was a

combination of cross-plume traverses and an along-plume segment in which the G-1 travelled against the wind, 45 km toward the fire line.

Time sequences of scattering and nr-PM1 measured on the along-plume flight segment are shown in Fig. 8a.  MSE, which is the ratio of these quantities is given in Fig. 8b as a continuous function and

averaged over each of 7 transects. The along-plume flight segment starts at 10:04:00 in dilute smoke. Though not readily apparent given the scale of the figure, a plume signature is seen in CO, scattering and nr-PM1 which are 360 ppb, 403 $Mm^{-1}$, and 114 $\mu g\ m^{-3}$, respectively, averaged over transect 7. At 10:06 there is a transition to a higher concentration plume region which encompasses transects 8-13.  Between 10:11:50 and 10:11:51, a plume boundary is crossed into much cleaner air.  Scattering and nr-PM1 increase

between 10:04:00 and 10:11:50, but not monotonically as the path of the G-1 is not consistently oriented along the path of a hypothetical, non-meandering point source plume.

From Fig. 8a, one can see that the ratio of scattering to nr-PM1 increases with downwind distance (decreasing clock time).  For the purpose of illustration, Fig. 8b presents 1 Hz values for MSE generated

from the smoothed data in Fig 8a (see figure caption).  Without smoothing, 1 Hz MSE is too noisy to be useful.  In order to bypass such difficulties, we determine transect-averages as in Eqs. 2 and 3.  According to the transect-average values in Fig. 8b, MSE increases from 1.9 to 3.6 as plume age changes from 0.12 to 1.0.



The along-plume flight segment of flight 821b provides an illustration of the time dependence of
aerosol neutralization, which we describe in terms of a ratio of equivalents: $(Cl^- + SO_4^{2-} + NO_3^-) / NH_4^+$.
This expression is a simplification as there are other basic anions and acidic cations besides these four
measured ions.  Also, these anions are not necessarily associated with $H^+$. Fig. 9, shows that near-source
aerosol has primary sulfate, nitrate, and chloride that is not matched by the uptake of $NH_3$ resulting in an
acidic ion balance over most of the plume. There is a steady trend towards neutralization with downwind
distance, with an equivalence ratio of one nearly reached at the furthermost downwind point of the along-
plume transect. For almost all of the along-plume segment, the number of equivalents of $NH_4^+$ is very
nearly the same as the number of equivalents of $NO_3^-$, despite the abundance of $SO_4^{2-}$ and $Cl^-$, suggesting
that much of the sulfate and chloride is in a non-acidic form (Akagi et al., 2012).


### *4.3 Aging in Wildfire Plumes*

Transect average quantities in excess of background, normalized for dilution, are used to determine
the evolution of aerosol concentration and optical properties as functions of photochemical age.  Quantities
considered in this section are $\Delta nr\text{-}PM1/\Delta CO$, $\Delta$light scattering/$\Delta CO$, MSE ($\Delta$scattering/$\Delta nr\text{-}PM1$ mass),
BC mass ratio ($\Delta BC/\Delta nr\text{-}PM1$), MAC ($\Delta$light absorption/$\Delta nr\text{-}PM1$ mass), and SSA
($\Delta$scattering/($\Delta$scattering+$\Delta$absorption).  The age dependence of these variables for each flight are shown in
Figs. 10 and 11.  In order to compare flights we have defined fresh emissions as having an Age of 0.2 and
aged emissions as having an Age of 1.0.  This range is spanned (or very nearly so) on 8 of 9 flights, the
exception being 822a with a lowest Age of 0.38. Linear least squares fits provide values at the fresh and
aged limits.  Comparisons between individual data points and the least squares fits in Figs. 10-11 show, in
general, that the fresh and aged points give a good representation of trends.  Properties of fresh and aged
emissions are collected in Table 5.  Changes due to aging are given by $(X_{Aged} - X_{Fresh})/X_{Fresh}$, in Table 6.

On the 9 pseudo-Lagrangian flights, aerosol mass normalized by CO, $\Delta nr\text{-}PM_1/\Delta CO$, varies
between a 22% increase and a 29% decrease.  The average change is a 12% decrease with a one σ standard
deviation (16%) that encompasses no net aerosol production.  Figure 10b shows scattering normalized to
CO as a function of photochemical age.  According to the summary in Table 6, the aging change in
normalized scattering ranges from constant to nearly doubling, with a 9-flight average equal to a 41%
increase.  As, on average, there is no increase in nr-aerosol mass with age, the normalized scattering
increase is not due to more aerosol; rather, it is due to a time evolution in aerosol microphysics.  Figure 10c
shows the time evolution of scattering per unit of nr-aerosol mass.  MSE increases with photochemical age
for all 9 flights; the average increase is 56% with a standard deviation of 20% and range 33 to 97%.

Figure 11c indicates that on 7 of 9 flights, absorption per unit mass of aerosol is either independent
of- or decreases slightly with age.  The increase observed for flight 809a appears to be within the scatter of



data points. A nearly constant absorption is consistent with results presented by Sedlacek et al (2018a) and much like the observations of Forrister et al (2015) which extend to longer time periods. Changes in SSA with atmospheric processing (Fig. 11d) are therefore controlled by the increase in scattering as the plume ages (Fig. 10c). The lowest SSAs observed were 0.8 to 0.85 in fresh smoke. As the plume ages for 2 – 3

hours, SSAs are 0.9 to 0.98.

### 4.4 Aerosol composition

Age related changes in the chemical composition of BB aerosol are presented here in terms of the fractional contribution of black carbon, organic compounds and inorganic ions ($\Delta NH_4^+$, $\Delta Cl^-$, $\Delta NO_3^-$. and

$\Delta SO_4^{2-}$) to transect-averaged $\Delta$nr-PM1. Organic compounds are characterized by the fraction of organic mass at m/z=43, 44. and 60, ($f43$,  $f44$, and $f60$ respectively) and by the elemental ratios O to C and H to C (O:C and H:C, respectively). Figures S9-S10 of the Supplement show these quantities as a function of photochemical age. In describing the organic aerosol, we make use of a body of work that shows that aerosol properties depend on values of $f44$ vs. $f43$ (e.g. triangle plots of Ng et al., 2011) and values of O:C

vs. H:C (e.g. Van Krevelen diagrams used by Heald et al., 2011 and Ng et al. 2011).

### 4.4.1 rBC and inorganics

Aerosol composition measurements for fresh and aged BB aerosol are summarized in Fig. 12. In every transect, more than 90% of aerosol mass is OA. Flight-averaged rBC is between 0.5 and 2.5% of nr-

PM1. On average, the mass fraction of rBC in aged smoke is 11% higher than in fresh emissions (Fig. 12). This increase is consistent with a constant amount of rBC as the concentration of nr-PM1 decreases by very nearly the same amount (12%, Table 6) relative to CO. Aging yields a change in $\Delta rBC/\Delta CO$ of 0.5%. There is a one-σ standard deviation of 21%, which is due to uncertainty of our methodology (i.e. fire inhomogeneities, uncertain backgrounds) and measurement uncertainties in CO and rBC. Inorganic species

constitute less than 10% of nr-PM1. Differences between fires are comparable to the differences between fresh and aged emissions. In the 730b case study $Cl^-$ differed by about an order of magnitude according to fuel type. It is likely that primary $Cl^-$ is in the form of non-volatile KCl and the decrease in $Cl^-$ reflects the formation and volatility of HCl (Akagi et al., 2012). On most flights $SO_4^{2-}$ decreases with age but the correlations are poor. An increase is expected from oxidation of $SO_2$ (Yokelson et al., 2009; Akagi et al,

2012). $SO_2$ emission factors measured on flights 730a and 730b are given by Liu et al (2016). $NO_3^-$ and $NH_4^+$ increase with age (Fig. S9). Increases in $NO_3^-$ would occur from the formation and partitioning of $HNO_3$ to the aerosol phase.

Fig. 12 shows that the 9-flight average acidity follows the aging pattern seen on flight 821b (Fig.

9b). Fresh emissions are acidic with $NO_3^-$ equivalent to $NH_4^+$, despite the presence of $SO_4^{2-}$ and $Cl^-$. The same caveats apply regarding a description of acidity in terms of the 4 ions readily quantified by the AMS.





### 4.4.2 Organic aerosol

The speciation of organic compounds is expected to vary with age as primary compounds are lost
by evaporation or transformed by aerosol phase chemistry. The later pathway includes the transformations
that convert primary organics to TBs (Adachi et al., 2019). As a plumes ages, gas phase oxidation of VOCs
creates less-volatile products which partition to the aerosol phase as SOA. Though organic mass is on
average constant in BBOP, the constituent species become more oxidized and viscous with age. Transect-
average H to C and O to C ratios (H:C, and O:C, respectively) observed in 9 wildfire flights are depicted by
a Van Krevelen plot in Fig. 13. Isolines of OSc, the average carbon oxidation state, defined to good
approximation as OSc = 2 O:C - H:C (Kroll et al., 2011), indicate, in each flight, an increase in carbon
oxidation state with photochemical age. For the 9 BBOP flights, slopes in the Van Krevelen plots vary
from near zero to -1 (average value = -0.4), implying flight to flight variability in the predominant
oxidation mechanism and products. For example, aerosol would age with a slope of zero if oxidation
proceeded by adding OH or OOH moieties; a slope of -1/2 by formation of a carboxylic acid with
fragmentation and a slope of -1 by the addition of a carboxylic acid without fragmentation. Adachi et al
(2019) used scanning transmission X-ray spectroscopy and electron energy loss spectrometry analysis to
show that the formation of TBs is accompanied by an increase in carboxylic acids.

A comparison with other BB aerosol elemental ratios (Ng et al., 2016), shows that BBOP elemental
ratios have values characteristic of semi-volatile OOA species and that the 2 to 3.5 hours of aging observed
on BBOP flights leads to composition changes in the direction of lower volatility species. A similar
conclusion that BBOP aerosol is in the semi-volatile category, progressing to lower volatility with age is
supported by values of *f*43 and *f*44 shown via a "triangle plot" in Fig. 14. The age dependence of *f*43, *f*44,
*f*60, H:C, O:C, and OSc is given explicitly in Fig. S10. Quantities directly related to aerosol oxidation state,
*f*44, O:C and OSc monotonically change with age for all flights, albeit with some scatter and flight-to-flight
differences. *f*60 which is a surrogate for primary levoglucosan, and related compounds (Cubison et al.,
2011) decrease with age for all flights (Fig. S10c).

Measurements from two regional scale BBOP flights, 806a and 816a, (not part of the current study)
have been compared with observations at MBO. According to back-trajectories, MBO sampled in the same
air masses as the G-1 after 6-12 hours additional processing (Collier et al., 2016; Zhou et al., 2017; Zhang
et al., 2018). A PMF analysis of the MBO observations (Zhang et al., 2017) indicated 3 BB components
that differed in the degree of chemical processing, The least processed component, BBOA-1 had an O:C
ratio = 0.35; the most processed component, BBOA-3, had an O:C ratio = 1.06, and was significantly less
volatile (up to a temperature of 200 °C) than the other PMF components. The aged BBOP samples have an
O:C ratio between 0.25 and 0.35 (Fig.13) and on that basis most resemble BBOA-1. It appears from the
age-trends in Fig. S10e that O:C continues to increase past the values observed in our 2 – 3 hour old





samples (i.e. there is no indication of O:C reaching an asymptote). Aged BBOP samples had significant

mass fractions of refractory TBs. Based on volatility, these aged BBOP samples resemble BBOA-3. The

difference between aged BBOP samples and aerosol measured at MBO indicate the importance of chemical

aging in the 3 to 15 hour time frame.

**Section 5 Increase in MSE with age**

**5.1 Observations**

Transect-average measurements in Fig. 10c and summarized in Table 6, indicate an average

increase in mass scattering efficiency (MSE) of 56% in aged as compared with fresh aerosol. For each

transect a surrogate for particle size is determined from FIMS measurements as follows: Size spectra are

first averaged over a transect. The peak in the transect-averaged size spectrum is determined from a local

quadratic fit that uses the 3 to 5 size bins nearest the largest value of dN/dLogDp. A restriction to Dp > 100

nm yields a peak for dN/dLogDp within the accumulation mode. We will refer to this diameter as $D_{GEO}$, in

analogy with the geometric mean of a log normal distribution.

Fig. 15 shows that $D_{GEO}$ is an increasing function of photochemical age in accord with multiple

field observations of BB aerosol (e.g. Akagi et al., 2012; Eck et al., 2013; Carrico et al., 2016). MSE is

observed to increase with $D_{GEO}$ (Fig. 16). This trend is expected for size distributions dominated by

particles smaller than the wavelength of scattered light, here 550 nm. Laing et al. (2016) observed a

correlation ($r^2$) between $D_{GEO}$ and MSE of 0.73 for an ensemble of 19 BB plumes impacting MBO in a one

month study. In BBOP, correlations between MSE and $D_{GEO}$ are high for individual flights (average $r^2$ =

0.75), but in contrast with the observations of Liang et al (2016), the ensemble of flights shows little

relation between MSE and $D_{GEO}$. The low correlation between MSE and $D_{GEO}$ for the BBOP 9-flight

ensemble appears to be caused by real fire-to-fire differences; said differences might be diminished by

additional atmospheric processing in the more aged smoke seen at MBO by Liang et al (2016).

In the size range of our samples, backscatter ratios and scattering Angstrom exponents decrease

with particle size (Selimovic et al., 2019). An anti-correlation between these intensive parameters and

MSE is expected and is observed (Figs. 17-18), both for individual flights and for the ensemble of 9 flights

that cover a range of MSE from 1.5 to 6.

An increase in MSE could be caused by an increase in the real part of the refractive index, $m_R$. In

biomass burn aerosol generated in the laboratory or sampled in the ambient atmosphere, $m_R$ is observed to

be ~ 1.50 to 1.60 (Levin et al., 2010; Adler et al., 2011) with values above 1.60 occurring in smoke that has

a higher fraction of BC than seen in BBOP. Based on literature values and Sedlacek et al (2018a), we

estimate that wildfire generated aerosol in BBOP has a refractive index, $m = 1.53 – 0.02i$. A change in $m_R$

from 1.53 to 1.60 during plume aging would result in a 17 to 24% increase in MSE according to Mie





calculations of aerosol with a log normal distribution in which $D_{GEO} = 200$ nm and $\sigma_G$ (geometric mean standard deviation) between 1.4 and 1.8. However, there are reports that $m_R$ is insensitive to aging (Levin et al., 2010) or decreases in aged aerosol (Adler et al., 2011).

*5.2 Coagulation Calculations*

An increase in MSE with age is more likely to be caused by a rearrangement of particle mass, favoring large diameter efficient scatters at the expense of small inefficient scatters. An example calculation shows that coagulation can lead to a substantial (factor of 2) increase in MSE.

Effects of coagulation upon an initial aerosol size distribution (Sakamoto et al., 2016) were determined using a Brownian coagulation kernel and algorithms in Jacobson et al (1994). Calculations were initialized with the near-fire FIMS particle size distribution measured on transect 13 of the 821b flight and integrated for 6600 sec, corresponding to the estimated atmospheric transport time on the along-plume flight segment, between the fire and the furthest downwind sample on transect 7 (see Fig. 7). Particles with

diameters above 260 nm are qualitatively accounted for by fitting the FIMS measurements to a double log normal that extends to 1000 nm. Based on CO measurements, the simulated air mass was diluted by a factor of 4 between transects 13 and 7. Particle concentrations in background air are negligible compared with the plume and are ignored. The dilution rate is an average value, so it is not expected that calculated concentration will follow the increases and decreases observed as the G-1 encountered more and less

concentrated areas within the plume. Errors in calculated aerosol concentration effect rates of coagulation but do not directly affect the aerosol size distribution shape or MSE which are intensive quantities.

Transect-averaged aerosol size distributions, extrapolated to 1000 nm, for the along-plume segment of flight 821a are given in Fig. 19 along with the corresponding size distributions determined from

coagulation calculations. The near-source size distributions have peaks in the Aitken and accumulation mode size range. The FIMS and calculated size distributions show a growth in the number of accumulation mode particles relative to the Aitken mode as well as a shift in the accumulation mode to larger particle diameters as time increases. Coagulation calculations produce changes comparable to the FIMS measurements between transects 13 and 10. Further downwind, the transfer of material from Aitken to

accumulation mode size ranges is more pronounced in the FIMS observations than in the coagulation calculations. MSEs calculated from the observed and calculated aerosol size distributions are compared in Fig. 20. Although, MSEs determined from FIMS size distributions and coagulation calculations are systematically lower than those observed (by factors of 0.83 and 0.64, respectively; see Fig. 20 caption), all three methods yield an MSE that increases by approximately a factor of two between the fire on transect 13

and the most downwind transect, 7.

*6. Discussion*



Correlations found in BBOP flights between i) MSE and ii) $D_{GEO}$, aerosol Angstrom exponent, and aerosol backscatter ratio are evidence that increases in MSE with time are due to the growth of aerosol

particles, such that they become more efficient scatters. Similar correlations have been found in other biomass burn studies (Levin et al., 2010; Liang et al., 2016; Selimovic et al., 2019). Coagulation was shown to be one mechanism for redistributing mass between different size particles that can lead to a doubling of MSE. Because the BBOP data set is restricted to aerosol size measurements below 210 or 260 nm, it is difficult to quantify whether coagulation alone can reproduce the observed MSE age dependence.


Coagulation is certainly not the only aerosol microphysical process occurring as evidenced by aerosol chemical composition changes with age (Fig. S10). Chemical changes in the organic component of BB aerosol have been interpreted in terms of mass transfer between particulate and gas phases. In this scenario, high volatility POA evaporates as the plume dilutes. Gas phase reactions of primary flame

emissions or the evaporated POA, creates oxygenated lower volatility VOCs that subsequently partition as SOA to the aerosol phase (Grieshop et al., 2009; Hennigan et al., 2012; Jolleys et al., 2012; May et al., 2013; Morgan et al., 2019). Transfer of material between the gas and (bulk) aerosol phase and amongst particles in the aerosol phase is driven by the thermodynamic constraint of equalizing chemical potentials. Though thermodynamics defines quasi-stationary states in an evolving plume, actual distributions are

commonly dictated by the kinetics of mass transfer within particles, between phases, and between individual particles (Marcolli et al., 2004; Zhang et al., 2012)

Time-dependent changes in the chemical composition of wildfire plumes observed in BBOP are consistent with a decrease in POA compensated by an increase in SOA, such that the total concentration of

organic aerosol, corrected for dilution, is approximately constant. An increase of SOA is inferred by the systematic increase in $f44$, O to C ratio and carbon oxidation state with time (Figs, S10b, e, and f). A decrease in $f60$ (Fig. S10c) has been linked to the evaporation of primary levoglucosan and related compounds.

Changes in chemical markers used to characterize the carbon oxidation state are several-fold smaller than observed in laboratory burns and in most field samples. Thus, the BBOP data set occupies a small fraction of composition space depicted in "triangle" diagrams of $f44$ vs $f43$ (Ng et al., 2011) and a similarly small fraction of a Van Krevelen diagram (Heald et al., 2010). Arrows that one could construct between our fresh and aged samples point towards the more aged compositions that others have observed

(se Figs. 1 and 4, Ng et al., 2011). Most of the plume samples described in this study have been exposed to atmospheric processing for two hours or less; the oldest sample are exposed for less than 3.5 hours. In contrast the data sets used in Heald et al. (2010) and Ng et al. (2011) contain both fresh and more processed BB smoke with atmospheric residence times (or equivalent OH exposure) of days.





Given that there are aged-related changes in organic composition (though relatively small) consistent with loss of POA and gain of SOA, is it possible that a transfer of mass between the gas and particulate phase and amongst particles can yield a size distribution in which MSE increases by as much as a factor of two?  It is likely that during mass exchange between the gas and aerosol phase, equilibrium amongst particles of different sizes is not maintained since the air mass is evolving rapidly (i.e. <1 hour)

and mass transfer between particles can have substantially longer time constants (Marcolli et al., 2004). The kinetics of diffusion in the continuum regime tends to favor evaporation of small particles since in that regime, $dDp/dt \sim Dp^{-1}$.  A preferential evaporation of small particles occurs also for the transition regime size particles that produce most of the scattering in wildfire plumes. As total mass is nearly invariant, the concurrent process of condensation of SOA cannot have the same kinetics as evaporation of POA, else

particles of a given size will have no net size change. If SOA was to be distributed amongst particles so as to equalize mole fractions (i.e. follow a volume growth law), then growth of large particles would be favored over smaller ones and MSE would increase.  Detailed calculations invoking a wider choice of evaporation and growth options as well as possible kinetic limitations caused by within-aerosol diffusion are required to quantify effects of mass transfer on MSE.


## 7. Conclusions

     As part of the BBOP campaign in the Pacific NW U,S., nine wildfire plumes were sampled in a pseudo-Lagrangian mode to determine the time evolution of pollutants between the fire and after up-to 3.5 hours of daytime atmospheric processing.  Atmospheric processing was quantified by a photochemical age

defined as $- Log_{10}(NO_x/NO_y)$.  Typical ages were between 0.2 and 1.0 and maximum downwind times close to 2 hours, though on occasion as long as 3.5 hours. Plume concentrations were corrected for dilution using CO as a conservative tracer.  Background subtraction depended on observations in near-by clean air.

     On average, normalized aerosol mass concentrations were constant over several hours of

atmospheric aging.  Mass scattering efficiency increased with age by an average of 56%; the range amongst flights was an increase of 33% to 97%.  Except for two flights, mass absorption coefficients (MACs) are in the high teens and nearly independent of age.  If absorption is due to coated BC, then coatings are formed early in the BB plume and of a thickness such that absorption is insensitive to further coating (Bond et al, 2006; Forrister et al., 2015).  Scattering, normalized for dilution, increases with age causing SSA to

likewise increase. In fresh smoke albedos were 0.85 – 0.90.  These SSAs increased to ~ 0.95 in aged smoke. If we suppose our plumes to be over a portion of the globe with average albedo (that average including oceans, etc.), then the wildfire plumes when first observed would have a near-zero direct radiative effect on the Earth's radiative balance and a cooling effect after 2 – 3 hours (Selimovic et al, 2019; Eck et al 2013).





735   For typical BB particles, scattering and MSE are increasing functions of particle size.  In order to establish that aerosol particles grow with age we relied on i) a decrease in aerosol backscatter ratio and Angstrom exponent with age (as expected from Mie theory) and ii) an increase with age of $D_{GEO}$, the diameter, within the accumulation mode, at which point dN/dLogDp has a maximum value. $D_{GEO}$ is best thought of as a surrogate of particle size as it is obtained from measurements with Dp < 210 or 260 nm,

740 short of covering the full range of optically active particles.  Almost all values for the surrogate measures of Dp increase with age.  Fire to fire variability of these metrics can be larger than the difference between fresh and aged emissions, which cautions against comparing fresh emissions from one fire with aged emissions from another.

745   A calculation from a high concentration along-plume flight segment indicates that coagulation can cause a factor of two increase in MSE by the transfer of mass from the Aitken mode to the larger accumulation mode. Further calculations constrained by aerosol size distributions that fully cover the size range that contributes most to scattering, are required to determine the importance of coagulation and to identify other mechanisms that cause MSE to increase with age.


   In newly emitted plumes $NH_4^+$ and $NO_3^-$ tended to have an equivalence ratio near unity, despite the presence of $Cl^-$ and $SO_4^{2-}$.  That $HNO_3$ and $HCl$ were not volatilized argues against the initial $SO_4^{2-}$ and $Cl^-$ being cations of strong acids. Primary $Cl^-$ varied between 0.2 and 2.5%, with the higher value associated with grass fuels. Aged plumes had, on average, a neutral equivalence ratio (anions / $NH_4^+$ near

755 unity).  The ratio, rBC/nr-PM1 varied amongst flights, with a range of 0.5% to 2.5%.  In all cases organic aerosol constituted > 90% of nr-PM1.

   Organic composition changed with age as POA evaporated and SOA condensed.  Loss of POA can be inferred from the age dependence of $f$60, a surrogate for primary emissions of levoglucosan and related

760 compounds.  Condensation of SOA is seen from the increase in $f$44 and O to C ratio in aged samples. Changes with age of these mass spectra fragments, though robust, are smaller than that observed in most laboratory and field studies of aerosol aging. This is expected as our aged samples were typically only2 to 3 hours downwind of their wildfire source.

765 Data Availability. Data from the BBOP field campaign have been archived at https://www.arm.gov /research/campaigns/aaf2013bbop.

Author Contributions. LIK and AJS III were co-PIs of BBOP responsible for selecting objectives and designing the field campaign. RY advised.  The primary responsibility for collecting data was as follows: PTI and SP2, AJS III; SP-AMS, TBO and JS; Trace gas, SRS; $CO_2$, MKD;  FIMS, JW; Electron

770 microscopy, KA and PB; Measurements at MBO, QZ, SC, and SZ;  Analysis of effects of dilution in BBOP



flights, ALH and JRP; Mie calculations, EL. All authors contributed towards the analysis of data contained herein and towards writing the manuscript.

Competing Interests. The authors declare no conflict of interest.

**Acknowledgement**

This research was performed under sponsorship of the U.S. DOE Office of Biological & Environmental Sciences (OBER) Atmospheric System Research Program (ASR) under contracts DE-SC0012704 (BNL) and DE-AC05-76RL01830 (JES, PNNL). The Pacific Northwest National Laboratory is operated for DOE by Battelle Memorial Institute. M. Dubey thanks ASR for support. K. Adachi thanks the

support of the Global Environment Research Fund of the Japanese Ministry of the Environment (2-1703 and 2-1403) and JSPS KAKENHI (grant number JP19H04259 and JP16K16188) P. R. Buseck acknowledges support from the Pacific Northwest National Lab (PNNL) and the DOE Atmospheric Radiation Measurement (ARM) Program under Research Subcontract #205689. T.B. Onasch acknowledges support from the DOE ARM program during BBOP and the DOE ASR program for BBOP

analysis (contract DE-SC0014287). J.R. Pierce and A.L. Hodshire acknowledge support from the U.S. NOAA, an Office of Science, Office of Atmospheric Chemistry, Carbon Cycle, and Climate Program, under the cooperative agreement award NA17OAR4310001; the U.S. NSF Atmospheric Chemistry program, under Grants AGS-1559607 and AGS-1950327; and the US Department of Energy's Atmospheric System Research, an Office of Science, Office of Biological and Environmental Research

program, under grant DE-SC0019000. R. Yokelson effort was supported by NASA grant NNX14AP45G to the University of Montana.

Researchers recognize the DOE Atmospheric Radiation Measurement (ARM) Climate Research program and facility for both the support to carry out the BBOP campaign and for use of the G-1 research

aircraft. The authors gratefully acknowledge the skill and safety ethos of the AAF (ARM Aerial Facility) pilots, flight staff, and instrument mentors. We gratefully acknowledge assistance from C. Wiedinmyer of NCAR in providing Western U.S. fire statistics.

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





**Tables**

Table 1.  Aerosol and trace-gas instruments on G-1 used in this study

| Measurement[§] | Instrument or method |
|---|---|
| particle size spectrum<br> Dp = 20 – 350 nm [§§] | FIMS  $\mathcal{D}$ |
| particle size spectrum<br> Dp = 0.5 – 50 µm | CAPS |
| particles (cm$^{-3}$), Dp > 2.5 nm | TSI CPC3025  $\mathcal{D}$ |
| particles (cm$^{-3}$), Dp >10 nm | TSI CPC3010  $\mathcal{D}$ |
| nr-aerosol mass | SP-AMS |
| rBC mass | SP2  $\mathcal{D}$ |
| 3λ light scattering | TSI 3263 nephelometer |
| light absorption | PTI |
| $CO_2$ | cavity ringdown spectroscopy |
| CO | resonance fluorescence |
| NO | chemiluminescence |
| $NO_2$ | photolysis yielding NO |
| $NO_y$ | 350 °C Mo converter |

§ Particle size spectra were also measured by UHSAS and PCASP particle probes mounted on wing pylons. Data from these probes were not used because of severe coincidence problems.

§§ Upper size limit during plume sampling = 210 nm for flights 809a and 822a; 260 nm for flights 25a, 726a, 730a, 730b, 814a, and 822b.

$\mathcal{D}$ Instrument sampled from dilution line.





Table 2. Synopsis of BBOP research flights in the Pacific Northwest

| Flight | Fire/Other venue | Missing data [1] | Lagrangian ✓ reason why not [2] |
|---|---|---|---|
| 715a | Portland | CO, FIMS, $NO_x$ | |
| 717a | Papoose, Pine Creek, Rough Creek | CO, FIMS, $NO_x$ | x CO |
| 719a | Ridge, Summit, Pine Creek | $CO_2$, FIMS, $NO_x$ | transport direction |
| 723a | Sunnyside | FIMS, SP-AMS | x SP-AMS |
| 723b | Sunnyside | FIMS, PTI, SP-AMS | x SP-AMS |
| 725a | Mile Marker 28 / MBO | PTI | ✓ |
| 726a | Mile Marker 28 | | ✓ |
| 730a | Colockum Tarps | | ✓ |
| 730b | Colockum Tarps | | ✓ |
| 806a | Douglas & Whiskey Complex / MBO | | regional smoke |
| 809a | Colockum Tarps | $CO_2$, $NO_y$ | ✓ |
| 813a | Pony Fire Complex | $CO_2$, $NO_y$, FIMS<210nm | ✓ |
| 814a | Pony Fire Complex | $CO_2$, FIMS | ✓ |
| 816a | Douglas Fire Complex / MBO | | regional smoke |
| 821a | Government Flats | | plume at low altitude |
| 821b | Government Flats | | ✓ |
| 822a | Gold Pan Complex | FIMS < 210 nm | ✓ |
| 830a | Portland, Seattle | PTI | |
| 905a | Spokane | PTI | |
| 909a | Portland, Seattle | PTI | |
| 911a | agricultural burn | | |

FIMS size range for plume sampling was 20 nm – 210 nm for flights 813a and 822a: 20 nm – 260 nm for flights 725a, 726a, 730a, 730b, 809a, and 821b.

[1] Only refers to instruments used in this study

[2] Excludes urban plumes and agricultural burn. x indicates a key measurement is missing



**Table 3.** Wildfires used for time evolution study

| Fire | Fuel | Fire duration m-dd | Acres burned thousands | Flights |
|------|------|------|------|------|
| Mile Marker 28 | Grasslands, shrub brush, timber, and timber litter | 7-24 to 8-05 | 26.1 | 725a, 726a |
| Colockum Tarps | Short grass (1 Foot), Timber grass understory and Hardwood litter | 7-27 to 8-18 | 80 | 730a, 730b, 809a |
| Pony Complex | Timber, brush, grass | 8-9 to 8-19 | 149 | 813a, 814a |
| Government Flats | Timber (grass and understory). Hardwoods, pine and fir. | 8-16 to 9-10 | 11.4 | 821b |
| Gold Pan Complex | Timber (litter and understory) Unburned mixed conifer stand, bug killed Douglas fir and Lodgepole pine | 7-16 to 10-3 | 43 | 822a |





**Table 4.** Lagrangian flights used for analyzing time evolution of wildfire emissions

| Flight | Time of day (UTC) | Transects | Photochemical age | Transect-average value | | Peak value | | |
| | | | | MCE[b] | Temperature (°C) | nr-PM1 ($\mu g\ m^{-3}$) | CO (ppm) | Scattering ($Mm^{-1}$) |
|---|---|---|---|---|---|---|---|---|
| 725a | 19:29 - 20:16 | 8 | 0.18 – 1.03 | 0.92 ± 0.01 | 16 ± 1 | 1375 | 3.63 | 3607 |
| 726a | 20:27 – 22:11 | 10 | 0.17 – 1.03 | 0.89 ± 0.02 | 17 ± 2 | 2429 | 2.64 | 5002 |
| 730a | 16:07 – 17:44 | 11 | 0.11 – 1.38 | 0.93 ± 0.007 | 12 ± 1 | 1706 | 3.94 | 3143 |
| 730b | 20:14 – 21:51 | 13 | 0.17 – 0.99 | 0.92 ± 0.005 | 12 ± 1 | 2973 | 5.08 | 3912 |
| 809a | 18:34 – 19:56 | 15 | 0.20 – 1.05 a | n/a | 14 ± 0.2 | 1121 | 0.81 | 1744 |
| 813a | 19:09 – 20:58 | 10 | 0.20 – 1.06 a | n/a | 6 ± 1 | 4534 | 7.41 | 8344 |
| 814a | 18:01 – 20:12 | 18 | 0.23 – 1.38 | n/a | 6 ± 1 | 3416 | 7.19 | 7191 |
| 821b | 21:23 – 22:11 | 16 | 0.11 – 1.02 | 0.92 ± 0.006 | 15 ± 0.5 | 12098 | 16.1 | 16043 |
| 822a | 18:58 – 20:29 | 10 | 0.37 – 1.25 | 0.87 ± 0.01 | 7 ± 1 | 1032 | 2.31 | 3066 |

a Photochemical age determined from downwind distance and 7-flight fitting function shown in Fig. 2.

b $\Delta CO_2 > 2$ ppm, (±1 σ for variation between transects)


**Table 5** Flight averages for fresh and aged transects§.

| Flight | Type | N | Δnr-PM1/ΔCO µg m$^{-3}$/ppm | ΔScat./ΔCO Mm$^{-1}$/ppm | MSE Mm$^{-1}$ /µg m$^{-3}$ | ΔBC/Δnr-PM1 | MAC Mm$^{-1}$ /µg m$^{-3}$ | SSA | MCE |
|---|---|---|---|---|---|---|---|---|---|
| 725a | fresh | 1 | 492 | 865 | 2.29 | 0.012 | | | 0.917 |
| | aged | 5 | 378 | 1229 | 3.32 | 0.013 | | | 0.917 |
| 726a | fresh | 1 | 384 | 1125 | 2.83 | 0.016 | 0.185 | 0.923 | 0.913 |
| | aged | 8 | 352 | 1544 | 4.40 | 0.021 | 0.323 | 0.933 | 0.886 |
| 730a | fresh | 6 | 352 | 790 | 2.31 | 0.016 | 0.267 | 0.894 | 0.929 |
| | aged | 2 | 318 | 1287 | 4.09 | 0.016 | 0.062 | 0.984 | 0.874 |
| 730b | fresh | 2 | 296 | 700 | 2.36 | 0.016 | 0.303 | 0.884 | 0.918 |
| | aged | 5 | 363 | 1218 | 3.40 | 0.015 | 0.239 | 0.930 | 0.917 |
| 809a | fresh | 2 | 325 | 936 | 3.16 | 0.024 | 0.264 | 0.914 | |
| | aged | 4 | 230 | 1074 | 4.68 | 0.026 | 0.581 | 0.898 | |
| 813a | fresh | 1 | 344 | 915 | 2.71 | 0.011 | 0.248 | 0.919 | |
| | aged | 5 | 310 | 1203 | 3.89 | 0.013 | 0.242 | 0.943 | |
| 814a | fresh | 1 | 331 | 1003 | 2.94 | 0.010 | 0.205 | 0.928 | |
| | aged | 7 | 240 | 1108 | 4.72 | 0.011 | 0.181 | 0.966 | |
| 821b | fresh | 6 | 449 | 841 | 1.87 | 0.006 | 0.365 | 0.843 | 0.922 |
| | aged | 2 | 446 | 1668 | 3.86 | 0.009 | 0.346 | 0.910 | 0.919 |
| 822a | fresh | 0 | 278 | 1048 | 3.62 | 0.008 | 0.125 | 0.964 | 0.894 |
| | aged | 2 | 219 | 1047 | 4.82 | 0.008 | 0.112 | 0.977 | 0.857 |
| Avg, | fresh | 2.2 | 361 | 914 | 2.68 | 0.013 | 0.245 | 0.909 | 0.915 |
| | aged | 4.4 | 317 | 1264 | 4.11 | 0.015 | 0.261 | 0.943 | 0.895 |

§ Properties of fresh and aged smoke are determined from linear least squares fit of transect data as a function of age, evaluated at age = 0.2 and 1.0, respectively. For the purpose of categorizing the number of transects in fresh and aged smoke, fresh smoke has a photochemical age lower than 0.3 and aged smoke has a photochemical age = 0.8 – 1.2. Each flight has equal weight in average.



Table 6 Percent change, $100\,(X_{Aged}-X_{Fresh})/X_{Fresh}$, between aged and fresh emissions.

| Flight | nr-PM1/CO $\mu g\ m^{-3}$ /ppm | Scat./CO $Mm^{-1}$ /ppm | MSE $Mm^{-1}/\ \mu g\ m^{-3}$ | BC/PM1 | MAC $Mm^{-1}/\ \mu g\ m^{-3}$ | SSA[§] | MCE |
|---|---|---|---|---|---|---|---|
| 725a | -23.3 | 42.1 | 45.1 | 5.9 | | | 0.2 |
| 726a | -8.2 | 37.3 | 55.2 | 33.9 | -21.1 | 1.1 | -3.0 |
| 730a | -9.7 | 62.9 | 76.9 | 2.8 | 44.9 | 10.1 | -5.9 |
| 730b | 22.4 | 74.0 | 44.4 | -5.1 | 13.1 | 5.1 | -0.2 |
| 809a | -29.1 | 14.8 | 47.8 | 7.9 | 44.2 | -1.8 | |
| 813a | -9.8 | 31.4 | 43.7 | 16.1 | 3.4 | 2.6 | |
| 814a | -27.4 | 10.5 | 60.2 | 10.4 | -10.6 | 4.1 | |
| 821b | -0.6 | 98.2 | 96.8 | 44.9 | -39.2 | 7.9 | -0.3 |
| 822a | -21.3 | -0.2 | 33.2 | 2.9 | 7.4 | 1.3 | -4.1 |
| Average | -11.9 | 41.2 | 55.9 | 13.3 | -9.3 | 3.8 | -2.2 |
| Standard deviation | 16.1 | 32.1 | 19.7 | 16.1 | 28.4 | 3.6 | 2.3 |

§ Percent changes in SSA are small because values of SSA are close to one.  Averaged over the 8 flights with data, aging causes the difference between the SSA of fresh smoke and unity to decrease by more than a factor of 2.





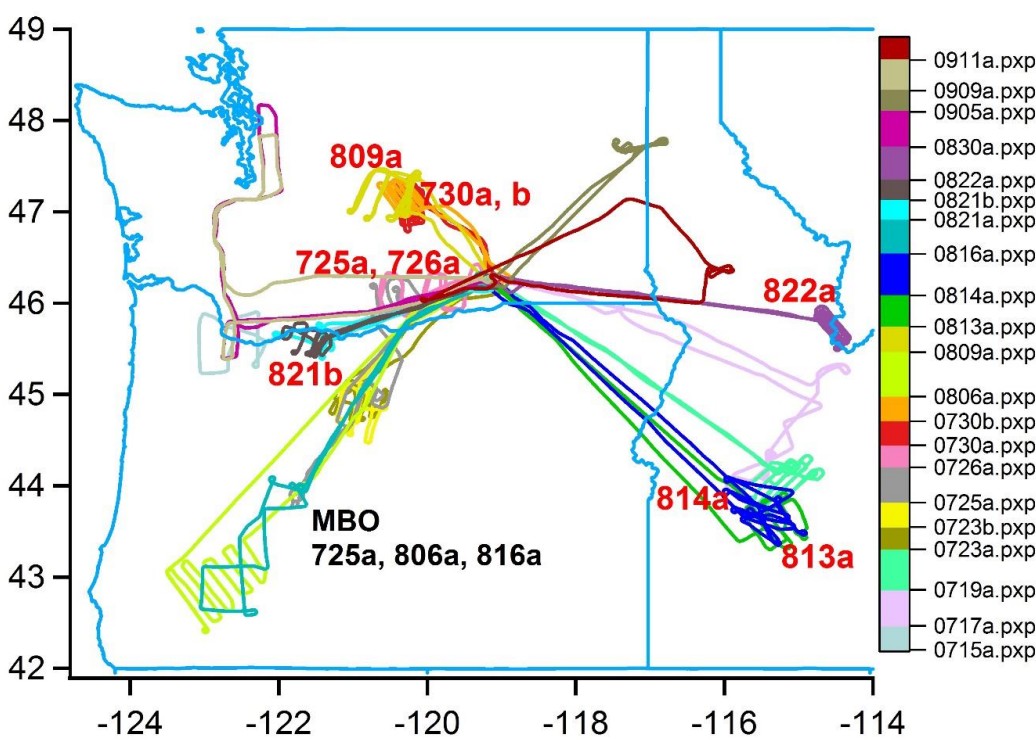

Figure 1. Ground track of G-1 aircraft for 21 BBOP flights in Pacific Northwest. Nine flights that were used in time evolution analysis are identified on map with red text.



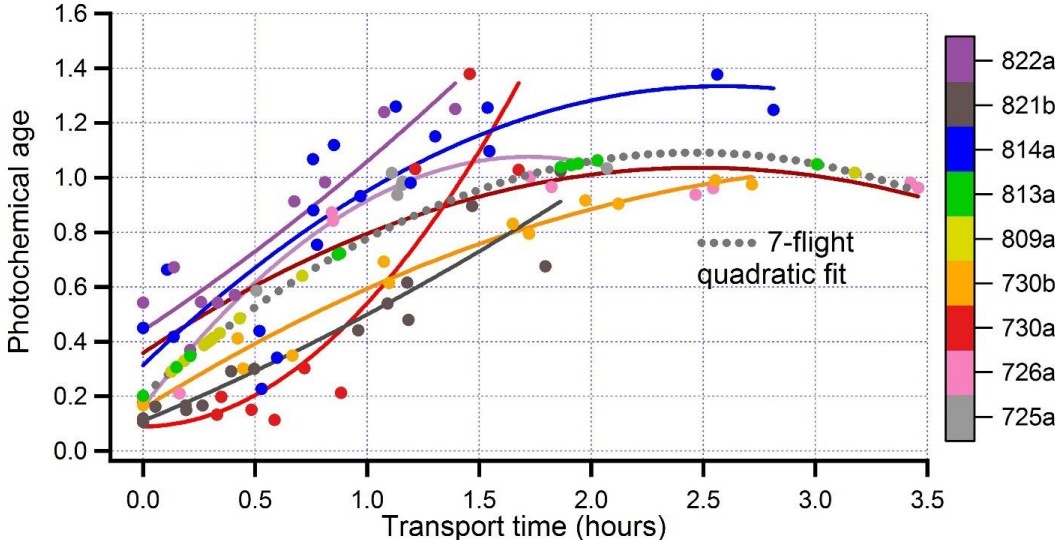

Figure 2. Comparison of photochemical age, - $Log_{10}$ ($NO_x/NO_y$), with downwind transport time calculated from distance and aircraft wind measurements. Each colored point is a transect of a flight identified in legend. Solid lines are quadratic fits to each of 7 flights. A quadratic fit for the combined 7-flight data set is used to estimate photochemical age for flights 809a and 813a, that are missing $NO_y$ measurements.





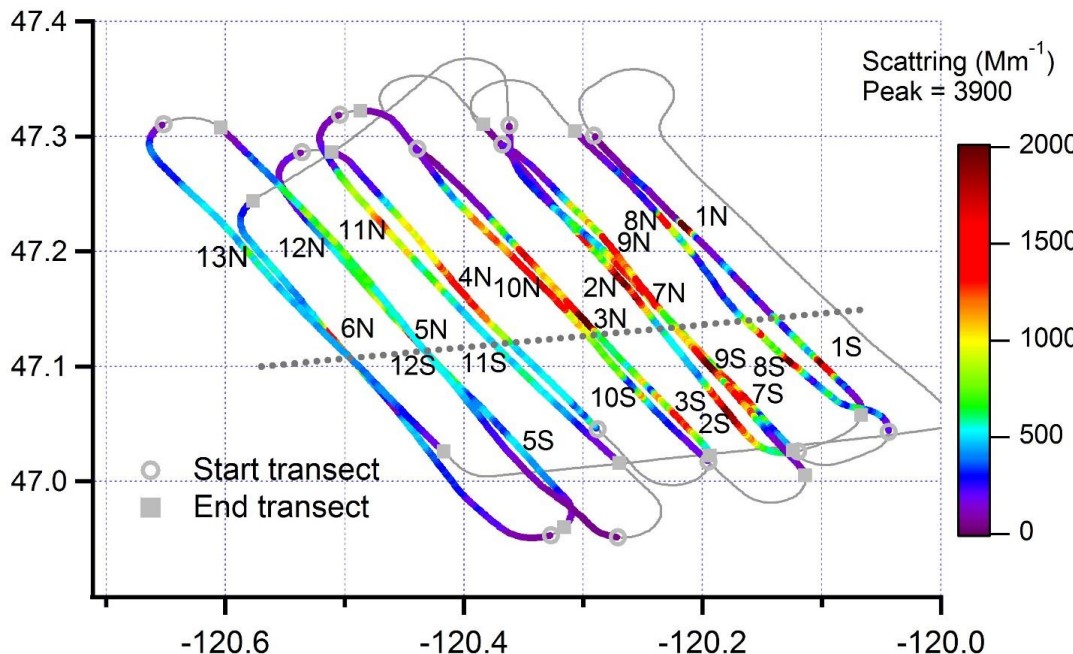

Figure 3. Ground track for flight 730b, colored by light scattering. The Colockum Tarp fire contained two plumes that have different ratios of chloride to total nr-PM1 (see Fig. 4a, following). Transects are labelled 1 – 13 in consecutive order and where possible the northern (low chloride) and southern (high chloride) regions are indicated by a N or S at the point of peak concentration (see Fig. 6, following). A dotted line approximately separates the two regions. Latitude and Longitude not to scale.





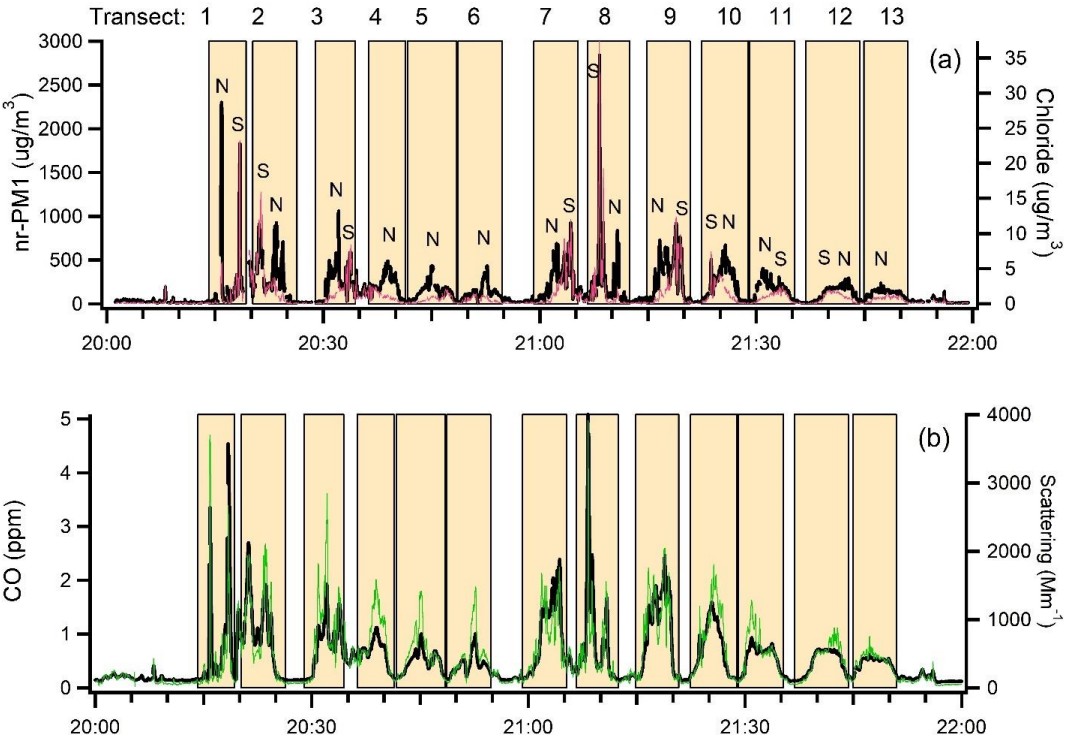

Figure 4 Time series data from flight 730b for (a) nr-PM1 and Cl⁻ (b) CO and light scattering. Data contributing to transect averages is identified by shaded rectangles. Transect numbers and North, South designation correspond to those in Figs. 3.





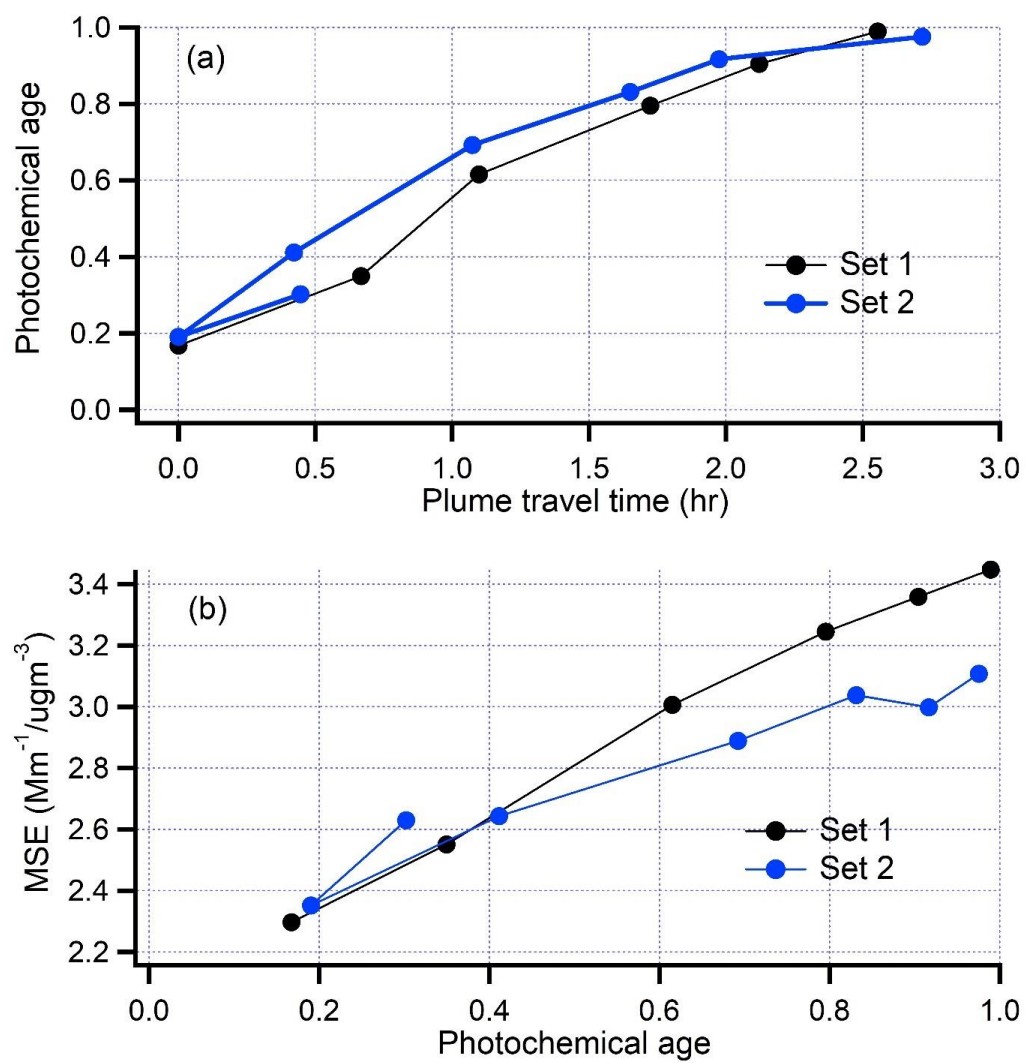

Figure 5 Transect average data from flight 730b. Lines indicate continuity in time of transects.





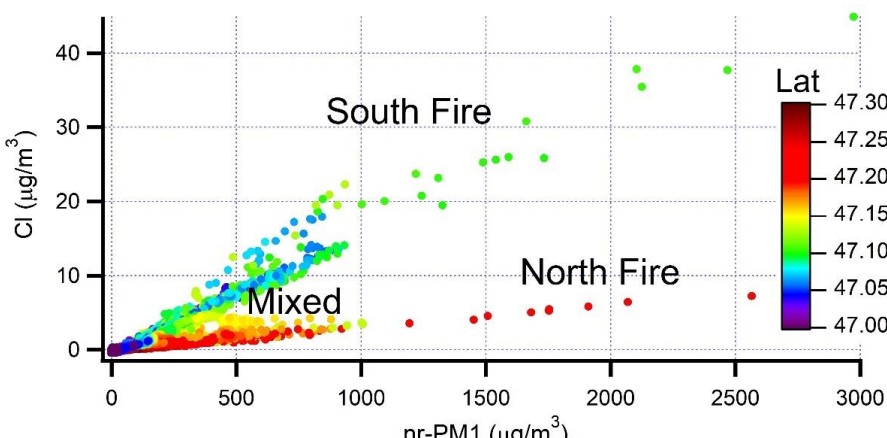

Figure 6. Cl⁻ concentration as a function of nr-PM1 for flight 730b. Data points are at 1 Hz. Spatial locations of North and South fires shown in Fig 3.

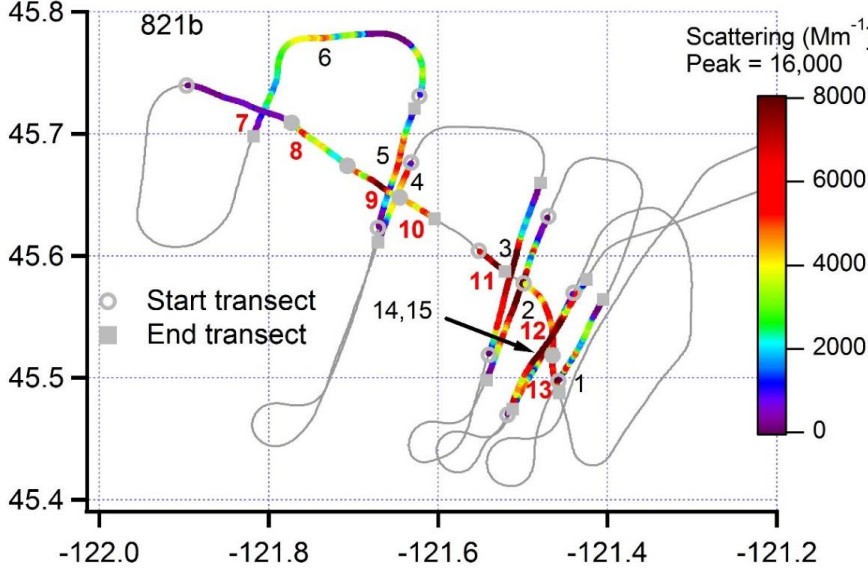

Figure 7. . Ground track for flight 821b with transects colored according to light scattering. Transects are labelled 1-15 in order of increasing flight time. Fire is near transect 1. Transect 7 is furthest downwind. Along-plume transects, 7-13, labelled in red. Transect 11 had limited $NO_x$ data and is not included in graphs in which photochemical age is the independent variable. Latitude and Longitude not to scale.





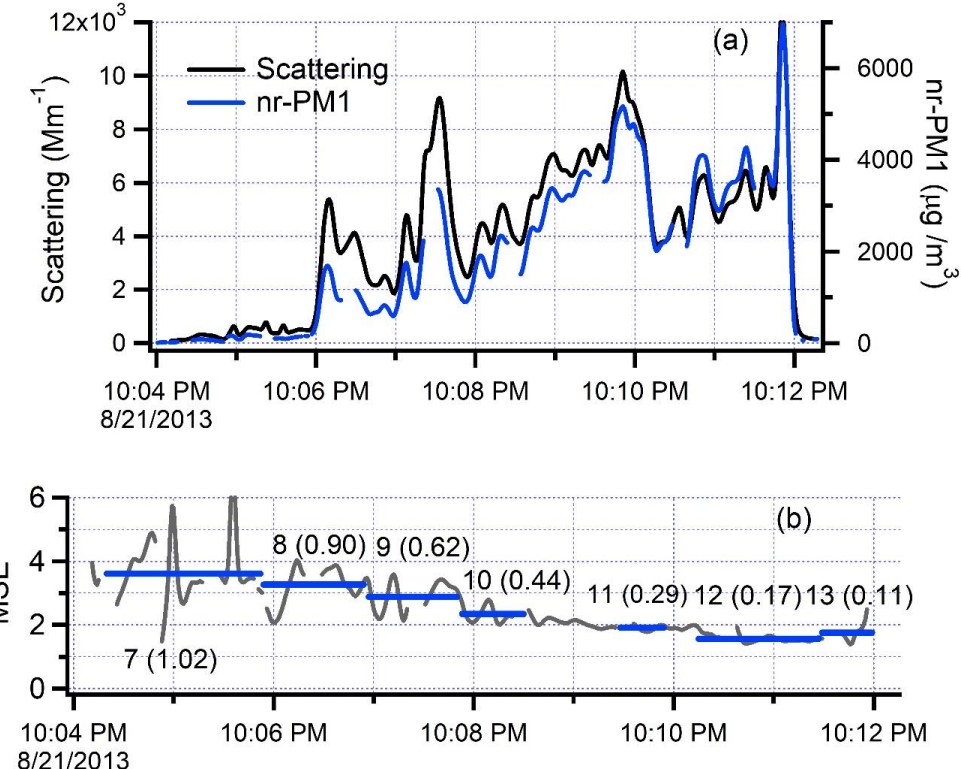

Figure 8. Time series from along-plume segment of flight 821b. The smoke plume is entered 45 km downwind of the fire at 20:04:30 UTC. Time increases to the right as the G-1 approaches the fire front, which is crossed at approximately 20:12 UTC. (a) Scattering and nr-PM1. Data from the nephelometer and SP-AMS have been smoothed with a 6 and 4 second binomial filter, respectively. (b) Continuous MSE (black trace) constructed from data in Fig. 8a. Blue lines are transect-average values. Labels refer to transect numbers shown in Fig. 7 with corresponding photochemical age in parentheses.



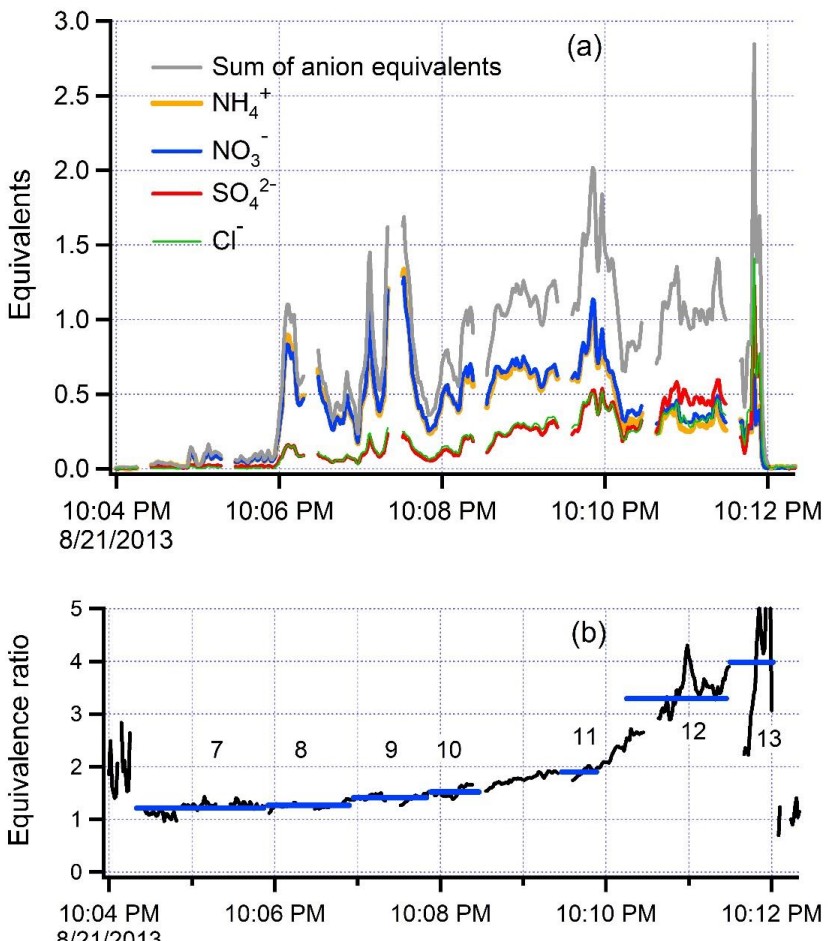

Figure 9. (a) Time series for inorganic species from the along-plume segment of flight 821b. Similar format as Fig. 8. Equivalents are equal to species molecular weight divided by charge. Sum of anion equivalents consists of $NO_3^-$ $SO_4^{2-}$. and $Cl^-$. (b) Equivalence ratio = ( $NO_3^-$ + $SO_4^{2-}$ + $Cl^-$ )/ $NH_4^+$. In the frame of the moving plume, the newly emitted smoke (transect 13) has an equivalence ratio of 4. The equivalence ratio steadily decreases nearly reaching a value of 1.0 at the end of the along-plume flight segment (transect 7). Values outside of the plume are at much lower concentration and susceptible to error from background subtraction .



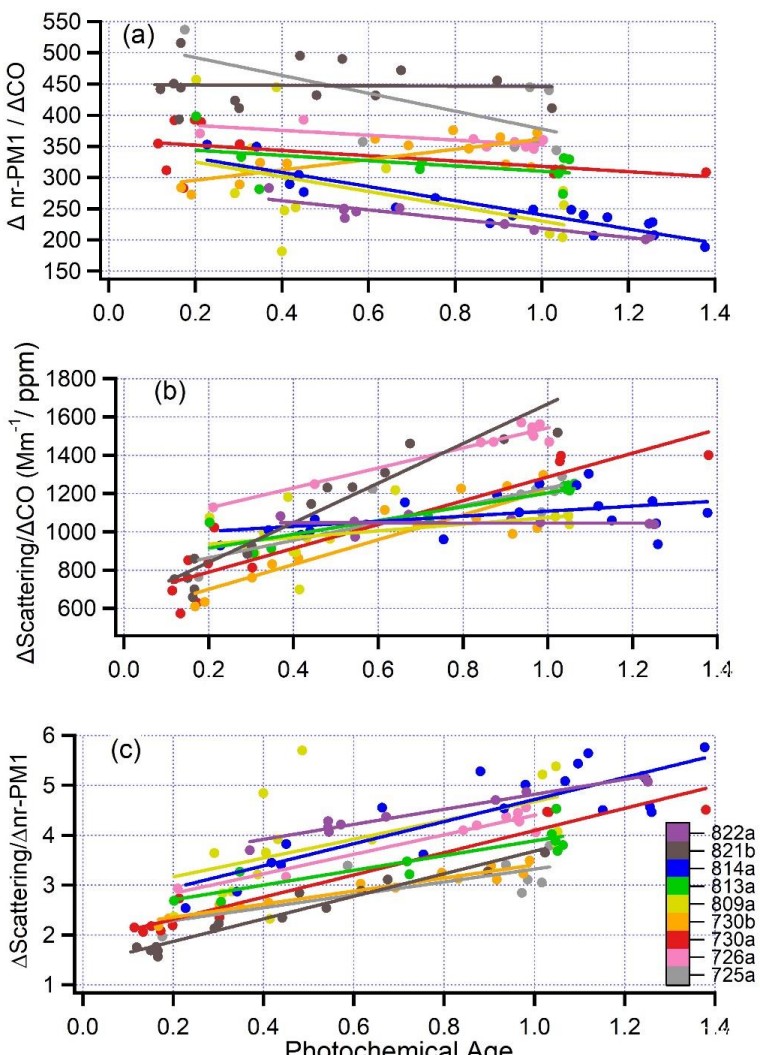

Figure 10. (a) nr-PM1, normalized by CO (µg/m³ aerosol per ppm CO) as a function of photochemical age for 9 pseudo-Lagrangian flights. Photochemical age for flights 809a and 813a has been determined from the relation between distance and age shown in Fig. 2. All variables are excess values with background subtracted. Each data point is a plume transect. Straight lines are ordinary least squares linear fits for each flight. (b) Scattering at 550 nm normalized by CO as a function of photochemical age for 9 flights. (c) Mass scattering efficiency (MSE) Scattering at 550 nm in Mm⁻¹ per µg/m³ nr-PM1 as a function of photochemical age for 9 flights.





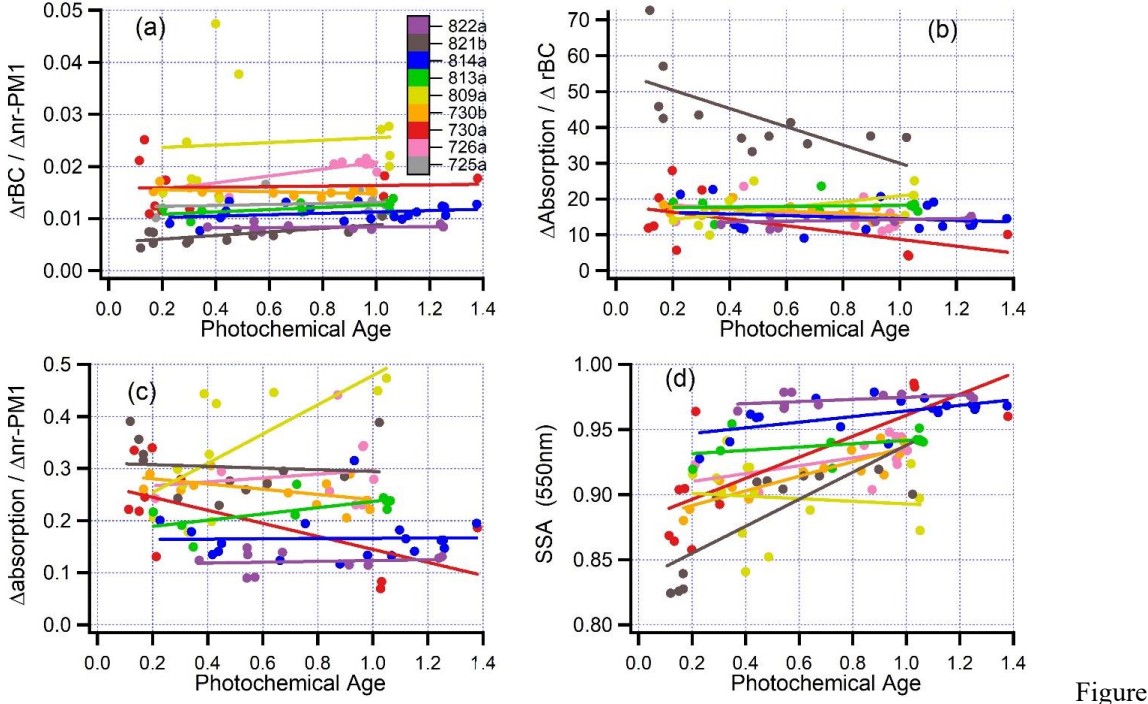

Figure 11. Ratios of excess plume variables as a function of photochemical age. (a) rBC/nr-PM1; (b) Absorption/rBC, $Mm^{-1}/(ug/m^3)$; (c) Absorption/nr-PM1, $Mm^{-1}/(ug/m^3)$; (d) single scatter albedo at 550 nm. Format same as Fig. 10.

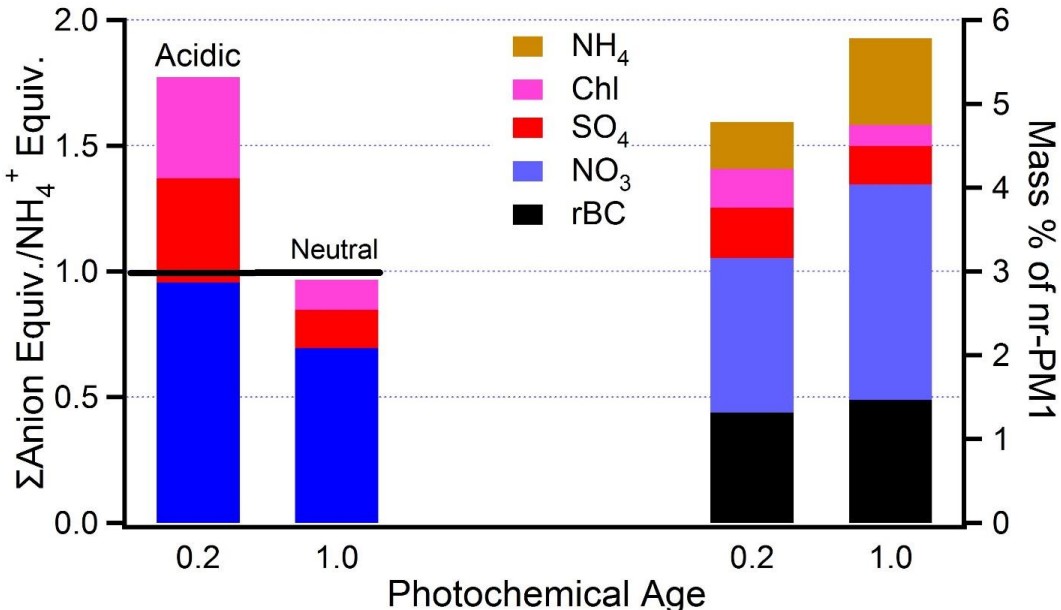

Figure 12. Inorganic composition of fresh and aged BB aerosol (photochemical age = 0.2 and 1.0, respectively) averaged over 9 flights. On left equivalence ratios of cations relative to $NH_4^+$. On right mass ratios relative to nr-PM1. The difference between the stacked bars on the right and 100% is organic aerosol.





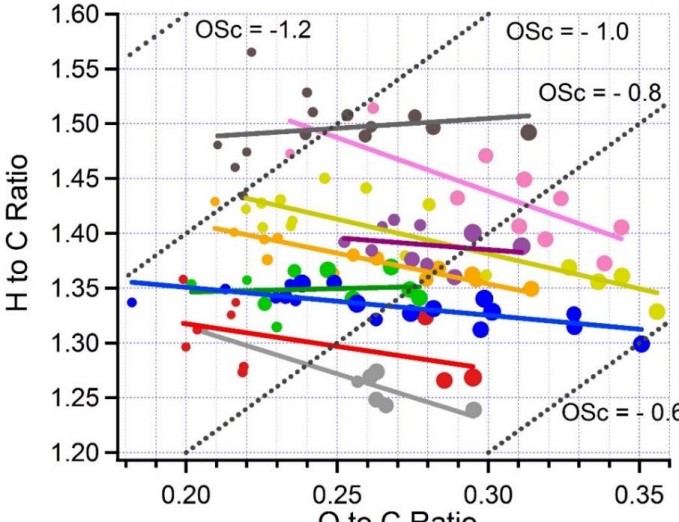

Figure 13. Van Krevelen diagram. Size of symbol linearly proportional to photochemical age. Dotted lines show H:C and O:C corresponding to labelled values of carbon oxidation state (OSc). Color code for flights same as in Fig. 11. Solid lines are linear least squares fit to data points in corresponding color.

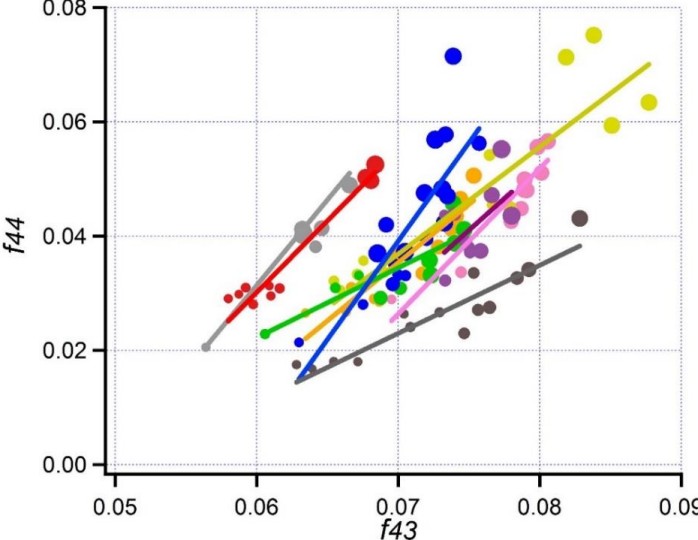

Figure 14. f44 vs f43 for transects on 9 pseudo-Lagrangian flights. Size of symbol linearly proportional to photochemical age. Same format as Fig. 13.

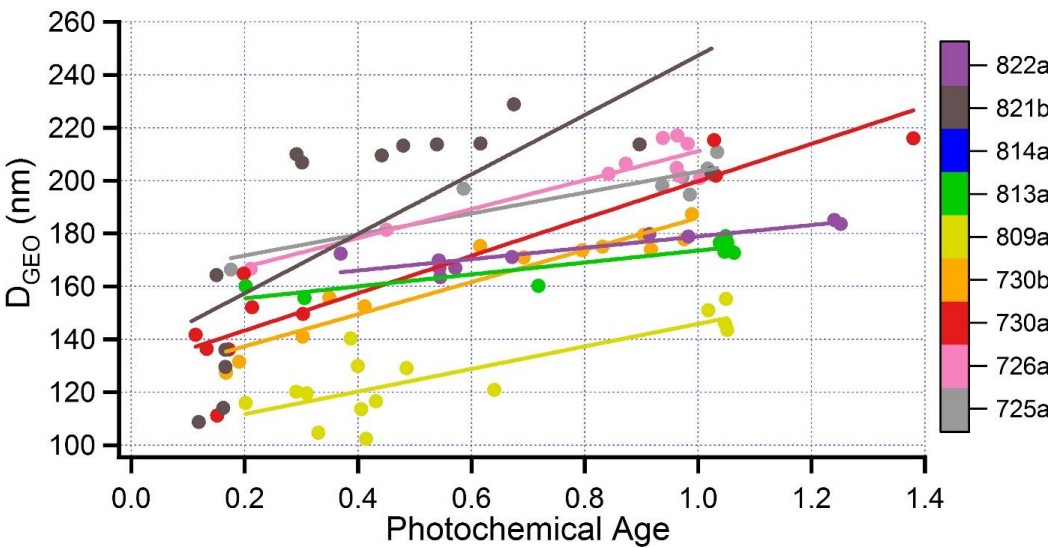

Figure 15. $D_{GEO}$ as a function of photochemical age for 8 flights. Calculation of $D_{GEO}$ from FIMS measurements given in text. Format similar to Fig. 10.

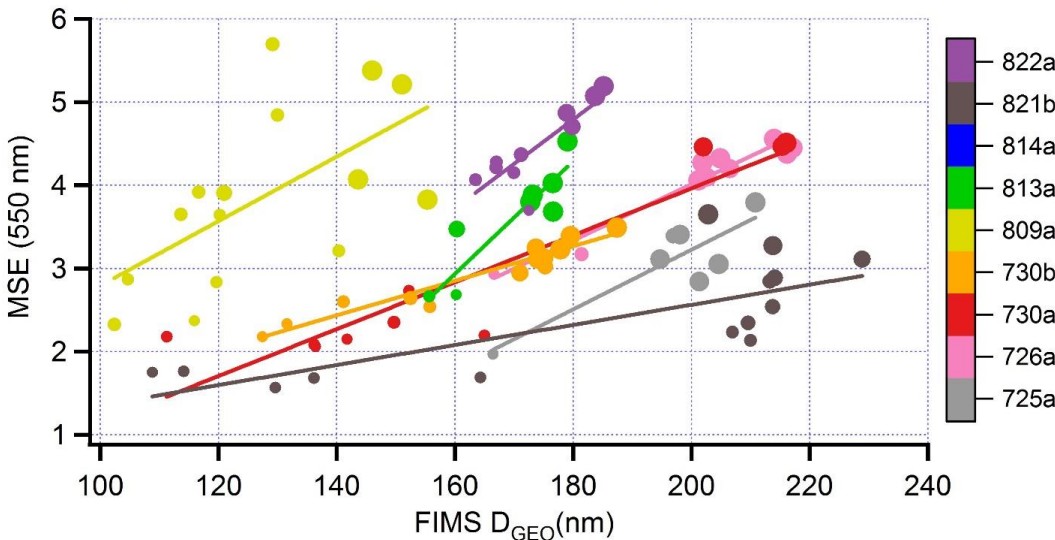

Figure 16. Mass scattering efficiency ($m^2/g$) as a function of $D_{GEO}$. Format similar to Fig. 13.





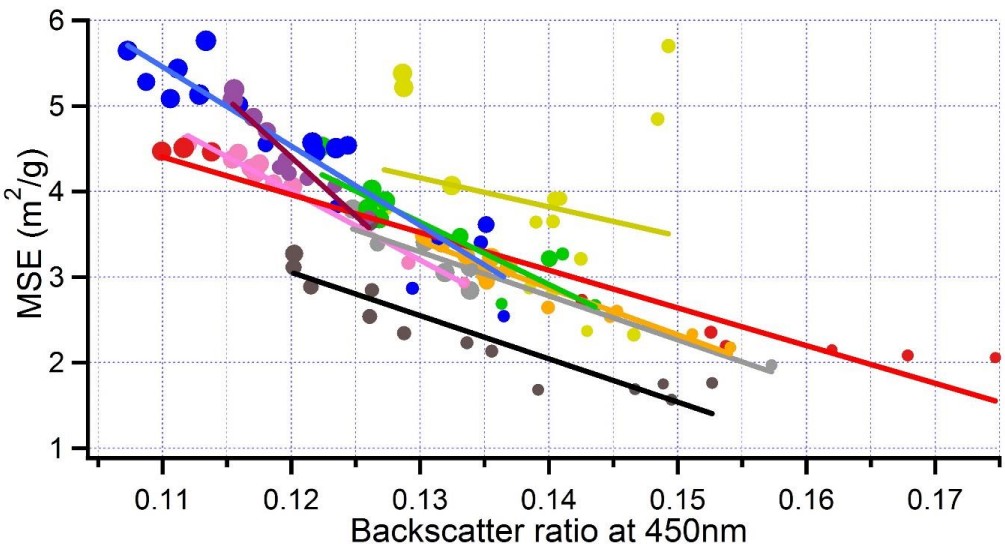

Figure 17. Mass scattering efficiency at 550 nm as a function of backscatter ratio at 450 nm. Same format as Fig. 13.

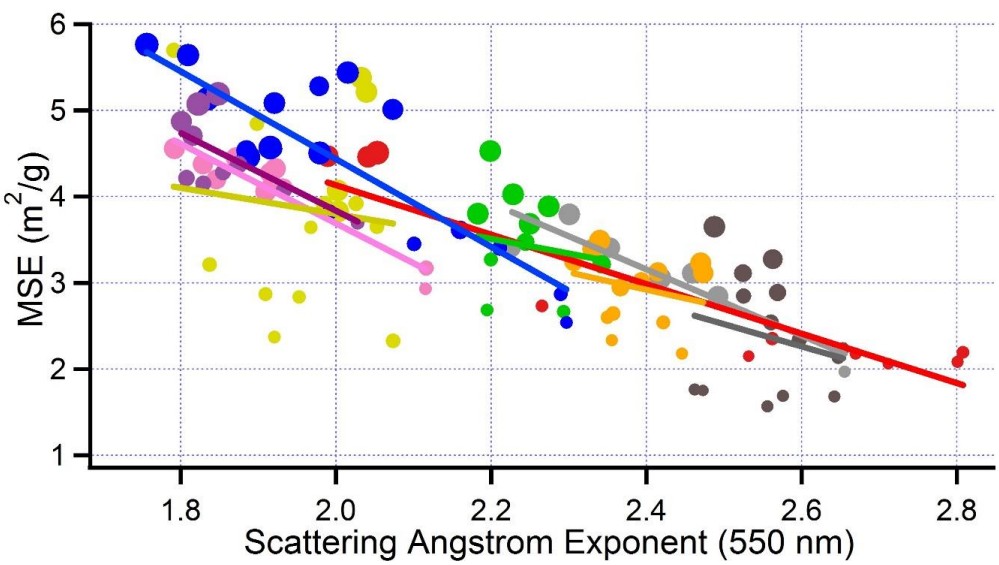

Figure 18. Mass scattering efficiency at 550 nm as a function of scattering Angstrom exponent at 550 nm. Same format as Fig. 13.



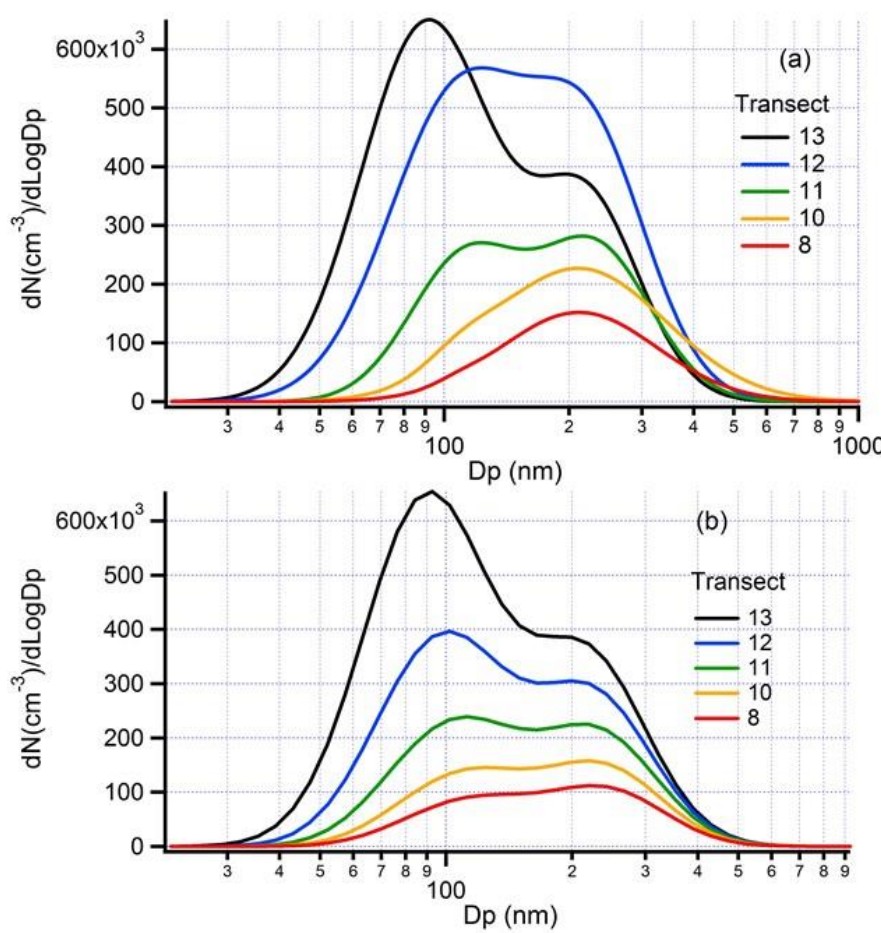

Figure 19.  Aerosol size distribution for 5 transects on along-plume segment of 821b (see Fig.7) (a) FIMS
data extrapolated to 1000 nm using a double log normal. (b) corresponding results of coagulation
calculation initiated with FIMS size distribution on transect 13.





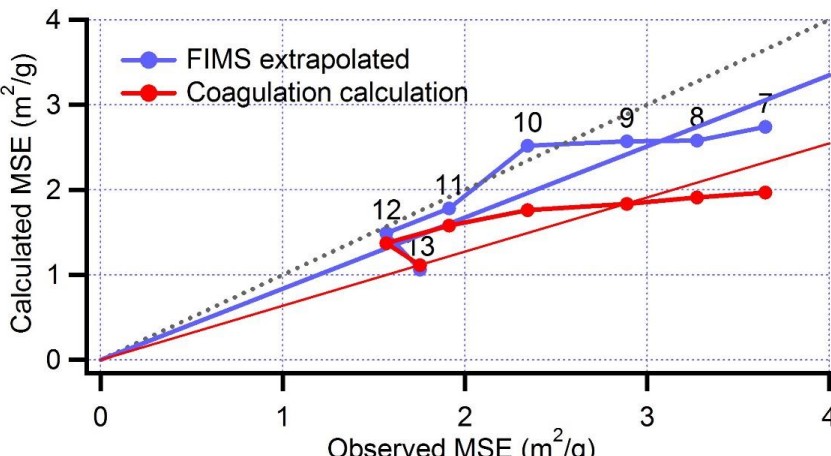

Figure 20. Mass scattering efficiency for transects 7 - 13 on the 821b along-plume flight segment (Fig.7). Mie calculations were used to determine MSE from size distributions in Fig. 19. Observed MSE is a ratio of scattering at 550 nm, to non-refractory mass measured via AMS. Blue and red lines with slopes 0.83 and 0.64 are linear least squares fit to data points. Corresponding $r^2 = 0.76$ and $0.77$, respectively. Dotted line is one to one.