# Peer review of "Rapid Evolution of Aerosol Particles and their Optical Properties Downwind of Wildfires in the Western U.S."

_Atmospheric Chemistry and Physics, 2020_

## Referee Comment (RC1) · Anonymous Referee #3 · 13 May 2020

Kleinman et al. presents measurements of aerosol collected during the BBOP field campaign in 2013. The analysis focuses on the evolution of aerosol downwind of the emission. The measurements range from a half hour to several hours old. At this timescale the authors identify the main trend as an increase in MSE along with other trends in aerosol composition and ion balance. These measurements represent an important airborne dataset sampling fresh wildfire smoke in the western US. The topic is appropriate for ACP and publication of these data and analysis is important in their own right and for comparison with other recent field campaigns (WECAN and FIREX-AQ) which also focused on sampling fresh wildfire smoke in western US.

[Figure]

The presentation of results this manuscript is complete but the discussion of the data is qualitative and is organized around short descriptions of the figures. Some panels in the figures are not mentioned in the text. Much of the results and discussion feel like a draft manuscript. I suggest that the author revise these sections.

Specific comments: Regarding the importance of coagulation in the aging of these plume, the manuscript oscillates between stating the coagulation is the main explanation for the increase in particle size and MSE downwind of the fires, and remaining agnostic about its contribution compared to evaporation/condensation. However, a coagulation calculation is presented for only one of the Lagrangian experiments (for the highest smoke concentration observed) and it suggested that coagulation can only account for a fraction of the increase in MSE. It seems that by performing the coagulation calculation for all of the plumes sampled here, the authors may be able to place a meaningful limit on the contribution of coagulation to the increase in MSE.

This same team used the BBOP dataset to analyze the formation of tarballs in these same plumes (Sedlacek et al. 2018). I expected to see some connection to tarballs in this analysis. Do the tarball sizes increase along with the rest of the aerosol? Are changes in the OA O:C ratio correlated with the tarball fraction? Some more connection would be informative as tarballs should account for significant fraction of the OA.

Technical Comments: Page 2 line 9 -11: Abstract states the main process is evaporation and condensation. Coagulation is not mentioned. implies disagreement with conclusions which are ambivalent about coagulation vs evaporation/condensation Page 2 line 11: typo 'transects' Page 7 ine: 203 'thermal denuder' -> 'thermodeunder' Page 12 line 375: missing a closing parenthesis Page 14 line 417: the sentence could much more precise Page 14 line 436. Missing period in et al. Page 14 lines 441-443: This sentence is confusing. Maybe the higher values come from hot spots (flaming spots?) in the burning area. Or are these separate smaller fire? Figure 3: scattering is misspelled. Show the location of the fire and an arrow showing the general wind direct would help orient the read more quickly. Longitude is capitalized in the caption Page

15 line 462 – 470: Please consider rewriting this paragraph. The reasoning is implied rather than explicit. The data discussed in the text (NOx, NOy , CPC) is not shown on the figure (scattering). CPC25 is not defined, although can be assumed to be from the TSI CPC3025 instrument. And that coagulation leads to a loss of small particles between the two transects. Page 15 line 465: CPC25 is not defined. Page 15 line 474: "and more extended in altitude." This seems likely but not discerned from the data presented. Figure 4: Consider coloring the bars for the two Lagrangian experiment different colors. Page 15 lines 475-476: To my eye, the divergence in between CO and scattering is more apparent in the North plume and the south plume. Maybe this should be discussed in the text/ or the north and south plumes could analyzed separately. Page 16 line 482: missing period Page 16 line 493: "Other Properties vary between North and South." Please expand a bit. What properties and how do they vary? Figure 5: Please add some more explanation of the figure to the caption. Page 16 lines 479- 484: For the second Lagrangian experiment, the transects are out of order. Please mention this in the text. Page 16 lines 504 -510: What was the purpose of transects 14 and 15? They are shown in figure 7, but not mentioned in the text. Transect 13 shows a kink in figure 20. I'm curious if transect 13 was not completely downwind of all of the emission for this fire and perhaps transect 12, 14, or 15 would be better for initializing the coagulation model. Figure 7: extra period in the caption. Please change the closing bracket to the parenthesis. Page 17 line 541 and 543: Age is unnecessarily capitalized. Repeated through the remained of the manuscript. Figure 11: Only panels c and d are briefly discussed in the text. Outliers such are simply explained away "as within the scatter". Panel b shows effective MAC, should one interpret the 821b flight to be heavy influenced by BrC as the MAC is much higher than expected from BC + lensing alone? This figure needs more detailed discussion in the text. Page 19 lines 590 -592: Is this also the case for flight 730a? It seems to be similarly outside the rest of the data. Page 21 line 667: typo, 'burn' should be 'burning' Figure 20: Is a linear fit to these data meaningful? Why are the intercepts held to zero? I would consider the transect 12 to be the initial transect, and constrain the linear fit to

go through transect 12. This interpretation of the figure suggests that coagulation can only account for a small fraction of the growth in MSE. The FINS MSE trends below the 1:1 line with increasing age for transects 10-7. Does this suggests that the FINS size distribution (even when extrapolated) is missing a significant fraction of the scattering or is this due to a changing index of refraction of OA as the sampling move downwind? Page 25 line 15: The coagulation calculation conclusion (factor of 2 increase MSE) rests on one data point from transect 13. If one initialed the calculation from transect 12, one might conclude a much smaller fraction of the MSE increase could be due to coagulation.
* * *

---

## Author Comment (AC1) · 22 Jul 2020

Kleinman et al. presents measurements of aerosol collected during the BBOP field campaign in 2013. The analysis focuses on the evolution of aerosol downwind of the emission. The measurements range from a half hour to several hours old. At this timescale the authors identify the main trend as an increase in MSE along with other trends in aerosol composition and ion balance. These measurements represent an important airborne dataset sampling fresh wildfire smoke in the western US. The topic is appropriate for ACP and publication of these data and analysis is important in their own right and for comparison with other recent field campaigns (WECAN and FIREXAQ) which also focused on sampling fresh wildfire smoke in western US. The presentation of results this manuscript is complete but the discussion of the data is qualitative and is organized around short descriptions of the figures. Some panels in the figures are not mentioned in the text. Much of the results and discussion feel like a draft manuscript. I suggest that the author revise these sections.

*Specific comments:* Regarding the importance of coagulation in the aging of these plume, the manuscript oscillates between stating the coagulation is the main explanation for the increase in particle size and MSE downwind of the fires, and remaining agnostic about its contribution compared to evaporation/condensation. However, a coagulation calculation is presented for only

one of the Lagrangian experiments (for the highest smoke concentration observed) and it suggested that coagulation can only account for a fraction of the increase in MSE. It seems that by performing the coagulation calculation for all of the plumes sampled here, the authors may be able to place a meaningful limit on the contribution of coagulation to the increase in MSE.

Your suggestion to look at the effects of coagulation upon size distributions and mass scattering efficiency (MSE) for the other 8 flights led me to find an error which reduces the role of coagulation by a factor of ~ 6 for 821b.  The coagulation calculations that yield the time dependent aerosol size spectra shown in Fig. 19a are correct, so any conclusion drawn from comparison with the observed (extrapolated) FIMS spectra in Fig. 19b still stand.  The Mie calculations were also correct.  Both programs have been checked against independent calculations.  However, the calculation of transect-averaged MSE was not correct.

Figure 20 has been removed. Coagulation is not the primary mechanism for changing size distributions and causing MSE to increase with photochemical age.  The coagulation calculations show that coagulation is an effective mechanism for transferring material from the Aitken to accumulation mode.

The text has been changed in several places.  Salient points are given below:

Abstract: "Coagulation is effective at moving aerosol from the Aitken to accumulation mode but yields only a minor increase in MSE"

5.2 Coagulation Calculations: "Observed and calculated size distributions show a growth in the number of accumulation mode particles relative to the Aitken mode as well as a shift in the accumulation mode to larger particle diameters as age increases.  The resulting increase in MSE was 17% for flight 821b and less (order 10%) for other flights. We assign a high uncertainty to these figures as mass and even more so, scattering, are primarily due to particles with diameters between the 260 nm upper limit of the FIMS and approximately 500 nm and are thus sensitive to the tail of our measurements and the method of extrapolation."

6.  Discussion:  "Coagulation moves mass from smaller to larger particles but its effect on MSE appears to be minor."

7. Conclusions:  "A calculation from a high concentration along-plume flight segment indicates that coagulation transfers particles from the Aitken mode to the larger accumulation mode, resulting in a small increase in MSE.  Further calculations constrained by aerosol size distributions that fully cover the size range that contributes most to scattering, are required to

verify the role of coagulation and to identify other mechanisms that cause MSE to increase with age."

This same team used the BBOP dataset to analyze the formation of tarballs in these same plumes (Sedlacek et al. 2018). I expected to see some connection to tarballs in this analysis. Do the tarball sizes increase along with the rest of the aerosol? Are changes in the OA O:C ratio correlated with the tarball fraction? Some more connection would be informative as tarballs should account for significant fraction of the OA.

Figure 6 of Sedlacek et al. (2018) shows tarball size, as measured on TEM grids, as a function of photochemical age. While the number of tarballs increases with age, there are not significant changes in their size distribution. Adachi et al (2019) presented electron microscopy results on chemical composition and size. Qualitative comparisons were made with AMS measurements of O to C ratio and FIMS measurements of particle size. Adachi et al. (2019) found that changes in organic aerosol O:C ratio are correlated with tarball fraction, but that finding (while comforting) does not provide quantitative information as both variables depend on photochemical aging. The key problem is that the in-situ instruments on the G-1 are designed to measure all particles, without bias. In contrast, as Adachi et al (2019) stated referring to electron microscopy "The vapor pressure of volatile and semivolatile organic and inorganic compounds in the sample chamber was too high to retain them in the particles (40, 59, 60), resulting in the loss of the volatile fraction from the particles collected on the TEM grids." In addition, particulate non-sulfate, O atom weight percentages (determined from a TEM operated in STEM mode) were obtained as a ratio to potassium. As demonstrated by Adachi et al. (2019), referencing composition data to K was not a concern in their analysis. It is, however, a concern if quantitative comparisons are made with AMS data, as the measurement of K with an AMS is problematic. In summary, there was little that we could add to add to material already published.

***Technical Comments:*** Page 2 line 9 -11: Abstract states the main process is evaporation and condensation. Coagulation is not mentioned. implies disagreement with conclusions which are ambivalent about coagulation vs evaporation/condensation

see coagulation discussion above

Page 2 line 11: typo 'transects'

Page 2 line 10, typo corrected

Page 7 line: 203 'thermal denuder' -> 'thermodeunder'

Page 6 line 184, changed

Page 12 line 375: missing a closing parenthesis

I cannot find the mis-matched parenthesis

Page 14 line 417: the sentence could much more precise

Page 12 line 381
Text changed to "As will be shown, $-\text{Log}_{10}\ (\Delta NO_x/\Delta NO_y)$ is a useful metric for chemical processing as it is strongly correlated with known age-related changes in aerosol composition due to atmospheric processing (see e.g. Fig. S10).

Page 14 line 436. Missing period in et al.

Changed in multiple locations

Page 14 lines 441-443: This sentence is confusing. Maybe the higher values come from hot spots (flaming spots?) in the burning area. Or are these separate smaller fire?

Page 12-13 line 404 – 406

MODIS FRE data often shows a pattern of non-contiguous pixels. Spaces between FRE pixels can be due to not plotting pixels below a threshold value, caused by poor visibility (i.e. clouds), or simply that there is no single point source or fire front.  For classification purposes these non-contiguous areas are identified as parts of a single fire. Given that the isolated "hot spots" are smaller than the main fire, the work of Hodshire et al (2019; 2020) shows a faster chemistry can be expected.  I have indicated that in the text as a possible cause of older smoke near the main plume.

Figure 3: scattering is misspelled.

Corrected

Show the location of the fire and an arrow showing the general wind direct would help orient the read more quickly.

Fire radiant energy from MODIS added to graphs for all flights except, 809a, for which it was not available. See also graphs in Hodshire et al., 2020 for plume trajectories, estimated by drawing a line through high concentration points at varying downwind distance.  We have wind

observations, but the flight path graphs are already very crowded.  Figure captions include information on plume direction and for some flights the transects closest and furthest from the fire.

Longitude is capitalized in the caption Page 15 line 462 – 470: Please consider rewriting this paragraph. The reasoning is implied rather than explicit. The data discussed in the text (NOx, NOy , CPC) is not shown on the figure (scattering). CPC25 is not defined, although can be assumed to be from the TSI CPC3025 instrument. And that coagulation leads to a loss of small particles between the two transects.

Page 13 line 424 – 431  "Longitude" in figure captions changed to lower case.

CPC25 was a typo.

In the rewriiten section the order of figures 4 and 5 are reversed.  Transects are labelled in (the new) Fig. 4.  New text includes:

"Highest aerosol and trace gas concentrations are observed on transects 1 and 8. There are short duration spikes in which ultra-fine particle concentration (determined from the difference between CPC3025 and CPC3010 measurements) are nearly $10^6$ $cm^{-3}$ and aerosol mass changes by more than 50% in one second.  Transects 2, 7, and 9 are 30 minutes downwind of transects 1 and 8.  The extra 30 minutes of aging allows coagulation to reduce the fraction of particles in the ultra-fine mode by more than a factor of 5, eliminates sharp gradients in concentration, and causes an increase in photochemical age from 0.2 to 0.35.  On that basis, it appears that transects 2, 7, and 8 are downwind of the main fire region.  However, MODIS indicates thermal anomalies downwind of transects 1 and 8 (see Fig. 3).  We cannot dismiss the possibility that the leading edge of the active burning region extends to nominal downwind transects, a consideration pertinent to other flights."

Ground track plots are all color-coded by scattering, with the scale chosen so that one can see the position of the smoke plume at varying downwind distances.  NOx/NOy and fine particle decrease markedly with distance and therefore would not serve my purpose.  I hesitate to add extra graphs.

Page 15 line 465: CPC25 is not defined.

Page 13 line 427; Typo, changed to CPC3025, which is identified in Table 1

Page 15 line 474: "and more extended in altitude." This seems likely but not discerned from the data presented. Figure 4: Consider coloring the bars for the two Lagrangian experiment different colors.

Page 13 line 435:       Deleted..

Page 15 lines 475-476: To my eye, the divergence in between CO and scattering is more apparent in the North plume and the south plume. Maybe this should be discussed in the text/ or the north and south plumes could analyzed separately.

Figure 5 takes up the better part of a page, yet it is hard to discern details.  I have restricted this figure to Transects 1 – 6 of Set 1 in the hope that the relation between scattering and CO can be more easily seen.  Text added to figure cation indicating that the downwind increase in scattering/CO is mainly a property of the low $Cl^-$ data.  The broader question is why not analyze the plumes separately.  There is merit to that approach, but it was decided that much of the data was from merged plumes and should, for the most part, be treated as such.

Page 16 line 482: missing period

Page 14 line 442:       corrected.

Page 16 line 493: "Other Properties vary between North and South." Please expand a bit. What properties and how do they vary?

Page 14 line 452:  A following sentence gives an example due to Onasch et al (2018) that the low $Cl^-$ plume has a higher O to C ratio (0.37 vs. 0.31).  Text added: The caption to Fig. 5 points out a higher scattering efficiency, relative to CO in the low $Cl^-$ plume.

Figure 5: Please add some more explanation of the figure to the caption.

Done. Figure is now Fig. 4. Transect numbers on graph will allow readers to visualize locations more easily.

Page16 lines 479- 484: For the second Lagrangian experiment, the transects are out of order. Please mention this in the text.

Transects are in time order.  At the end of the along-plume segment, after transect 13, the G-1 crossed the plume on transect 14, reversed direction and crossed  again on transect 15.  Text added to caption.

Page 16 lines 504 -510: What was the purpose of transects 14 and 15? They are shown in figure 7, but not mentioned in the text.

Data points for transects 14 and 15 are included in all graphs that use transect-average data from flight 821b, except for the figures that focus on the along-plume segment.  Data from transects 14 and 15 are consistent with along-plume measurements at comparable downwind distances. Transects 14 and 15 are located between - and roughly perpendicular to - transects 12 and 13. The MSEs determined for the sequence [12, 14, 15, 13] is [1.57, 1.68, 1.69, 1.75] m$^2$/g.  In general, cross plume transects contain the concentrated core as well as more dilute plume edges, whilst the along-plume transect is more heavily weighted towards the plume-core values. Comparisons between plume cores and plume edges for BBOP flights are described by Hodshire et al., Dilution impacts on smoke ageing: Evidence in BBOP data,  Atmos. Chem. Phys. Discuss., https://doi.org/10.5194/acp-2020-300, 2020 (2020)

Transect 13 shows a kink in figure 20. I'm curious if transect 13 was not completely downwind of all of the emission for this fire and perhaps transect 12, 14, or 15 would be better for initializing the coagulation model.

Our only real time indicator is downward pointing video,  The smoke obsures the ground till the end of transect 13.   We cannot tell from satellite imagery whether any oart of the fire  is downind of Transect 13.  Transect 13 shows a more pronounced Aitken mode than 12, 14, or 15, characteristic of the near-souce environment.

Figure 7: extra period in the caption. Please change the closing bracket to the parenthesis.
 Done

Page 17 line 541 and 543: Age is unnecessarily capitalized. Repeated through the remained of the manuscript.

Corrected

Figure 11: Only panels c and d are briefly discussed in the text. Outliers such are simply explained away "as within the scatter". Panel b shows effective MAC, should one interpret the 821b flight to be heavy influenced by BrC as the MAC is much higher than expected from BC + lensing alone? This figure needs more detailed discussion in the text. Page 19 lines 590 -592: Is this also the case for flight 730a? It seems to be similarly outside the rest of the data.

Discussions of absorption are being reserved for a future publication. You are correct that the 821b MACs have high values, especially near the source. I have cited in the paper satellite measurements of SSA from Noyes et al (2020) that show excellant agreement with the 821b in-situ measurements in Fig. 11C. The satellite measurements are based on data collected 2 hours before our in-situ measurements. In our paper, scattering gets most of the attention, but it would be mis-leading to not present absorption in the same format. Panel (a) shows rBC as a fraction of nr-PM1, an average of which appears in Fig. 12. The magnitude and time trend of rBC are discussed in Sec. 4.4.1. On average, the time trend for rBC is the same as for CO in agreement with expectations for conservative species. I have removed mention of outliers.

Page 21 line 667: typo, 'burn' should be 'burning'

corrected

Figure 20: Is a linear fit to these data meaningful? Why are the intercepts held to zero? I would consider the transect 12 to be the initial transect, and constrain the linear fit to go through transect 12. This interpretation of the figure suggests that coagulation can only account for a small fraction of the growth in MSE. The FINS MSE trends below the 1:1 line with increasing age for transects 10-7. Does this suggest that the FINS size distribution (even when extrapolated) is missing a significant fraction of the scattering or is this due to a changing index of refraction of OA as the sampling move downwind?

The FIMS does not cover particles larger than Dp= 260 nm, which contribute most of the scattering and volume. There is a clear underprediction of scattering and mass before the extrapolation. With the extrapolation, scattering and mass increase but not in a way that fortuitously gives an MSE that matches observations. If the actual size distribution above, say, 150 nm is a log normal, the extrapolation is unable to give the correct mode size and geometric standard deviation. It is also possible that a log normal is not the correct shape for the size distribution. Figure 20 has been removed.

Page 25 line 15: The coagulation calculation conclusion (factor of 2 increase MSE) rests on one data point from transect 13. If one initialed the calculation from transect 12, one might conclude a much smaller fraction of the MSE increase could be due to coagulation.

Figure 20 has been removed.

---

## Referee Comment (RC2) · Anonymous Referee #4 · 30 Jul 2020

Review of Rapid Evolution of Aerosol Particles and their Optical Properties Downwind of Wildfires in the Western U.S. by Kleinman et al., 2020, submitted to ACP

Kleinman et al. presented analysis on aging of biomass burning (BB) particles measured from BBOP field campaign (phase 1 mostly). The authors reported increasing mass scattering efficiencies and particle size with photochemical age downwind of the fire emissions. The authors concluded in the original manuscript that coagulation is the main mechanism for growing particles' size with aging. However, in the response to Review #3, the authors found mistake in the original manuscript and concluded that other mechanisms are responsible for the size change. It would be easier to review and com-

ment the revised manuscript. In general, the paper is well written and presents valuable datasets for the community to study BB particles. I find the manuscript suitable for ACP. Some concerns are attached below.

Line 16, one would think absorption by Brown carbon decays with time. SSA is quite sensitive to absorption instead of scattering. Some discussions and quantifications are needed to justify this statement.

After reading the review notes from Reviewer#3 and the authors' response, I am concerned with the calculation of coagulation vs. other microphysical processes in the paper. Seems the new calculation (not available to us yet) suggests that coagulation is not the primary contributor to growing size, then what are the primary mechanism(s)? Condensation? Aging? I understand the authors claimed in the response to Reviewer#3 that observations size limit is only up to 260 nm, thus difficult to tell exactly. However, seems the paper is incomplete without explaining the growing size, which leads to the increasing MSE (shift from 0 to cooling effect, the major finding of the paper in my point of view). If observation-based calculation is not easy, maybe some box model, regional model or theorical calculations can help.

Coagulation is mainly responsible for the size change from Aiken to Accumulation mode. How much does MSE change in response to the coagulation?

Any change in particle shape with aging available from BBOP? Need to consider (at least discuss) the non-spherical shape and its impact on scattering/absorption (SSA)?

Minor: a. Figure 4, labels of the left/right Y-axes should be in different colors

---

## Author Response (AR1)

Review of Rapid Evolution of Aerosol Particles and their Optical Properties Downwind of Wildfires in the Western U.S. by Kleinman et al., 2020, submitted to ACP

Kleinman et al. presented analysis on aging of biomass burning (BB) particles measured from BBOP field campaign (phase 1 mostly). The authors reported increasing mass scattering efficiencies and particle size with photochemical age downwind of the fire emissions. The authors concluded in the original manuscript that coagulation is the main mechanism for growing particles' size with aging. However, in the response to Review #3, the authors found mistake in the original manuscript and concluded that other mechanisms are responsible for the size change. It would be easier to review and comment the revised manuscript.

Response in blue

I agree. I uploaded a pdf as my response to the first review. The Copernicus upload menu requests that revised manuscripts not also be uploaded.

In general, the paper is well written and presents valuable datasets for the community to study BB particles. I find the manuscript suitable for ACP.

On behalf of my co-authors, thank you..

Some concerns are attached below.
Line 16, one would think absorption by Brown carbon decays with time. SSA is quite sensitive to absorption instead of scattering. Some discussions and quantifications are needed to justify this statement.

The omission of a discussion of brown carbon is intentional. My co-author and co-PI for BBOP, Arthur Sedlacek, is working on this problem. He has participated in field campaigns that cover a range of spatial scales with BBOP being near-field.

After reading the review notes from Reviewer#3 and the authors' response, I am concerned with the calculation of coagulation vs. other microphysical processes in the paper. Seems the new calculation (not available to us yet) suggests that coagulation is not the primary contributor to growing size, then what are the primary mechanism(s)?

The original coagulation calculations, shown in Fig. 19, are correct and that figure remains in the revised paper. The error was made in the calculation of MSE. I calculated MSE as a function of particle diameter, MSE(Dp). I integrated dN/dDp over Dp using MSE(Dp) as kernal, then normalized the result. The correct procedure is to determine scattering and mass, each as an integral over the size distribution, then divide one by the other. My reply to the first reviewer was that Fig. 19, showing the time evolution of the aerosol size distribution was correct but Fig. 20 showing MSE was incorrect and has been removed. I included in my reply the revised result that for flight 821b, the effect coagulation on MSE was reduced by a factor of 6, which works out to revised value of 17%. The draft manuscript was revised; the 17% value given.

Condensation? Aging? I understand the authors claimed in the response to Reviewer#3 that observations size limit is only up to 260 nm (the size limit for the FIMS is given in the original manuscript), thus difficult to tell exactly. However, seems the paper is incomplete without explaining the growing size, which leads to the increasing MSE (shift from 0 to cooling effect, the major finding of the paper in my pointof view). If observation-based calculation is not easy, maybe some box model, regional model or theorical calculations can help.

I agree that the paper emphasizes that the biomass burn smoke from wildfires we observed during BBOP undergoes a rapid increase in MSE with impacts on single scatter albedo, from which radiative changes can be inferred. This finding is derived from measurements of aerosol scattering and mass; it is independent of calculations. At the next level of understanding, we show that an increase in MSE is related to an increase in particle size, and decreases in aerosol Angstrom exponent and backscatter ratio. If I had not brought up coagulation as an explanation, one might still desire an explanation, but I doubt that the article would appear incomplete without one. There are any number of studies that are mainly observational as opposed to mechanistic. Often authors elect to parse a study into observational papers followed by calculation-based mechanistic papers.

I have some familiarity with the calculation of aerosol – gas phase interactions (Kleinman et al., 2009), which happened to follow an observational study (Kleinman et al., 2008). My opinion is that the BBOP data set does not contain the requisite measurements to define the mechanism(s) by which particle size increases, whilst total mass remains constant. In seminars, I have called this the Reverse Robinhood effect because the rich (large) particles increase in size by taking mass from numerous small particles. Meteorologists have worked on the (outwardly) similar process of drizzle formation in warm clouds for decades with - as far as I know - no definitive answers. Of course, a different set of mechanisms apply. The parallel is that there are observational papers, e.g. quantifying homogeneous and inhomogeneous entrainment as well as theoretical studies, e.g., relating mixing to turbulence..
.

As I mention in the Discussion section, within a BB plume there are gas phase oxidation reactions, evaporation of POA, and condensation of SOA.  Adding to the degree of difficulty is that there may be significant barriers to solid state diffusion (see Sedlacek et al.,2018) for the prevalence of tar balls in aged smoke, and Adachi et al (2019) for the physical properties of the BBOP tar balls.  The most that I foresee from the BBOP data set is a lengthy set of calculations that might or might not shed light on plausible mechanisms that could increase MSE.

Our manuscript quantifies phenomena that, to my knowledge, have not been previously described (e.g., relations between inorganic species in BB plumes, and small changes in aerosol oxidation state despite an order magnitude increase in photochemical age). The set of near-fire Lagrangian measurements in large wildfires in the western US is sufficiently sparse that our contribution is a significant part of the total.

Coagulation is mainly responsible for the size change from Aiken to Accumulation mode. How much does MSE change in response to the coagulation?

I'm not sure that I understand the question.  Coagulation is responsible for a 17% increase in MSE on flight 821b.  The 17% figure is included in a revised Sec. 5.2.  In case you are asking for the response of each mode, I have repeated the coagulation calculation using as initial conditions either the Aitken mode or the accumulation mode. Figure 1 below shows the double log normal fit to the FIMS data.  Figure 2 shows the effects of coagulation. Figure 2a is identical to Fig, 19 of the manuscript except that the transects are labelled according to downwind time,  The initial condition for (a) is the double log normal fit; for (b) a log normal fit to the Aitken mode; and for (c) a log normal fit to the accumulation mode.  Figure 3 shows the time evolution of  MSE. Results are given for the 2 mode case and for the Aitken and accumulation mode. The sum of MSEs for the Aitken and accumulation mode does not and should not be equal to the MSE for the double log normal  The 2 log normal solution shows a greater fractional increase in MSE as compared with the accumulation mode, reflecting the role of Aitken mode particles.  Its an interesting exercise, but does not show the way towards explaining the observed MSE trends.

Any change in particle shape with aging available from BBOP? Need to consider (at least discuss) the non-spherical shape and its impact on scattering/absorption (SSA)?

TEM images show particle composition and shape (Adachi et al., 2019). The electron microscopy is done under vacuum which evaporates volatiles.  To first order, images are consistent with spherical particles, many of which have been deformed and flattened upon impact. The exception are tarballs which are identified by their retention of a spherical shape.   . No evidence for crustal components is presented.  According to SP2 and AMS data, rBC is present at 1-2% of sub-micrometer aerosol, independent of age.  If there are any marine sources of NaCl, they are swamped by the overwhelming organic content of BB smoke.  K salts are

found by TEM as small inclusions.  A mass fraction is not presented.  On the basis of an age-independent ratio of K to CO (Fig S6 of Adachi et al.2019), it is argued that the loss of non-refractory components of primary particles is minimal  Overall, it appears that aerosol components that could lead to non-spherical components are small and have mass fractions that don't vary with age.  To first order it appears reasonable to ignore them.

Text added in 2.1.4 Scattering:
Given the measured aerosol composition, we assume that particles are spherical and that changes in scattering with respect to plume transport time is not caused by a change in particle shape.

Minor: a. Figure 4, labels of the left/right Y-axes should be in different colors

The old Fig. 4 is now Fig. 5.  I have changed color of right axis.

Adachi, K., Sedlacek, A. J., Kleinman, L., Springston, S. R., Wang, J., Chand, D., Hubbe, J. M., Shilling, J. E., Onasch, T. B., Kinase, T., Sakata, K., Takahashi, Y., and Buseck, P. R.: Spherical tarball particles form through rapid chemical and physical changes of organic matter in biomass-burning smoke, Proceedings of the National Academy of Sciences, 116, 19336, 10.1073/pnas.1900129116, (2019).

Kleinman, L. I., Springston, S. R., Wang, J., Daum, P. H., Lee, Y.-N., Nunnermacker, L. J., Senum, G. I., Weinstein-Lloyd, J., Alexander, M. L., Hubbe, J., Ortega, J., Zaveri, R. A., Canagaratna, M. R., and Jayne, J. The time evolution of aerosol size distribution over the Mexico City plateau. *Atmos. Chem. Phys*., 9, 4261-4278 (2009).

Kleinman, L. I., Springston, S. R., Daum, P. H., Lee, Y.-N., Nunnermacker, L. J., Senum, G. I., Wang, J., Weinstein-Lloyd, J., Alexander, M. L., Hubbe, J., Ortega, J., Canagaratna, M. R., and Jayne, J. The time evolution of aerosol composition over the Mexico City plateau. *Atmos. Chem. Phys*. 8, 1559-1575 (2008).

Sedlacek III, A.J., Buseck, P.R., Adachi, A., Onasch, T.B., Springston, S.R., and Kleinman, L., Formation and evolution of tar balls from northwestern US wildfires, *Atmos. Chem. Phys.*, 18, 11289–11301 (2018).

[Figure]

Fig 1. FIMS dN/dLogD averaged over transect 13, showing double log normal fit, and log normal fits to Aitken mode, and accumulation mode.

[Figure]

Figure 2. Time evolution due to coagulation of aerosol size distributions in Fig. 1 for (a) two modes, (b) Aitken mode, and (c) accumulation mode. Curves at 0, 900, 1800. 3600, and 5700 s, correspond to the initial conditions on transect 13 and the calculated size distributions for downwind transects 12, 11, 10, and 8 (see Fig. 19b of acpd submission)

[Figure]

Figure 3.  Time evolution of MSE.  Transect 8 is 5700 s downwind of transect 13, the starting point for the coagulation calculation.  A four-fold dilution occurs between time=0 and time=5700.

[revised manuscript text omitted]

---

## Author Response (AR2)

RE: Kleinman et al., acp-2020-239

Response to Editor regarding technical corrections

(1) Line 54, changed to Earth's

(2) Figure 10, caption corrected.